



# A history of TOPMODEL

Keith J. Beven[1], Rob Lamb[2], Mike J. Kirkby[3] and Jim E. Freer[4]

[1] Lancaster Environment Centre, Lancaster University, Lancaster UK

[2] JBA Trust, Broughton, UK and Lancaster Environment Centre, Lancaster University, Lancaster UK

[3] School of Geography, University of Leeds, Leeds UK

[4] School of Geographical Sciences, University of Bristol, UK and University of Saskatchewan, Coldwater Laboratory,

Canmore, Canada.

*Correspondence to:* Keith Beven, k.beven@lancaster.ac.uk

**Abstract.** The theory that forms the basis of Topmodel was first outlined by Mike Kirkby some 45 years ago. This paper
recalls some of the early developments; the rejection of the first journal paper; the early days of digital terrain analysis; model
calibration and validation; the various criticisms of the simplifying assumptions; and the relaxation of those assumptions in
the dynamic forms of Topmodel. A final section addresses the question of what might be done now in seeking a simple,
parametrically parsimonious model of hillslope and small catchment processes if we were starting again.

## 1.   Topmodel: the background

Topmodel is a rainfall-runoff model that has its origins in the recognition of the dynamic nature of runoff contributing areas
in the 1960s and 1970s that had been revealed in the data analysis of partial area contributions of Betson (1964) in Tennessee,
USA, and the field experience of Dunne and Black (1970) in Vermont, USA, and Weyman (1970, 1973) in the Mendips, UK.
It was one of the very first models to make explicit use of topographic data in the model formulation, hence the name of the
model. This was, however, well before digital terrain/elevation maps started to be made available[1]. The theory of Topmodel
aimed to reflect the way in which the topography of a catchment would shape the dynamic process responses and particularly
runoff generation on a variable contributing area. It did so in a structurally, parametrically, and computationally parsimonious

---

[1] It was also before the term "top model" started to be used in the fashion industry, though Ezio Todini was quick to have
some fun with the name at EGU when the first Italian issue of Topmodel magazine was published.



which gave it advantages over the full implementation of the physically-based model blueprint set out by Freeze and Harlan (1969).

The story of Topmodel starts when Mike Kirkby (MK) was at the University of Bristol where he worked with his PhD student Darrel Weyman in the East Twin catchment in the Mendips.   One critical observation from Darrel  Weyman's work was the
synchronicity of flows in a throughflow trough and in the main channel, suggesting the possibility that runoff per unit area might be approximately spatially constant; a key underlying assumption of Topmodel.    This analysis of the response of the upper East Twin led to the concept of a topographic index (as a/tanβ, a as upslope contributing area per unit contour length and tanβ  as local slope).

The first theoretical statement of Topmodel was presented in Kirkby (1975).   He wrote there: "*Any model with only a few parameters must necessarily simplify the spatial variation of moisture content over a drainage basin.   For a given average moisture content, there is a wide range of possible spatial distributions, even if rainfall is always spatially uniform, as is assumed here.   To predict the spatial consequences of an average moisture level some assumptions must be made about the duration of the rainfall inputs.   The simplest, which is adopted here, is to assume a time-independent steady state of net rainfall*
*input, ī* " (p.81).

Then, using the original nomenclature of Kirkby (1975), at any point, downslope flow per unit contour length, $q$, will be given by  $a(\bar{\imath} - q_o)$ where $a$ is the upslope contributing area to that point, and $q_o$ is a constant rate of leakage to the subsoil.   Then making the further assumption that the local hydraulic gradient can be approximated by the slope angle, $\tan \beta$, and the local
transmissivity can be represented as  $KS$ where $K$ is a permeability and $S$ is the local storage in rainfall equivalent depth units, then

$$q =  a(\bar{\imath} - q_o) = KS \tan \beta \qquad [1]$$

or

$$S =  a(\bar{\imath} - q_o)/KS \tan \beta \qquad [2]$$

This allows the condition for the soil to be just saturated to be defined as

$$S > S_o, or \ \frac{a}{K \tan \beta} > \frac{S_o}{(\bar{\imath} - q_o)} \qquad [3]$$


In terms of water balance accounting for the catchment as a whole, it is useful to integrate the expression for $S$ to provide a catchment average value $\bar{S}$.





$$\bar{S} = \frac{\lambda(\bar{i}-q_o)}{\bar{K}} \qquad [4]$$

where

$$\lambda = \frac{\bar{K}}{A} \int \frac{a}{K \tan\beta} dA$$

Combining these equations gives a condition for soil saturation in terms of the topographic index $a/\tan\beta$ where


$$\frac{a}{\tan\beta} > \frac{\lambda K S_o}{\bar{K}\bar{S}} \qquad [5]$$

The topographic index can be mapped in a catchment area as a function of the topography and then gives an indication of where a saturated contributing area might occur, and then how it might spread as a function of storage (e.g. figure 1 for the upper East Twin). The expression is simplified further if the permeability can be considered spatially constant and $\lambda$ simplifies

to the mean value of the topographic index in the catchment. The topographic index was also used later to compare with the saturated areas at Tom Dunne's Sleepers River field site in Vermont in Kirkby (1978) (figure 2). Kirkby (1975) also provides relationships for the leakage term $q_o$ and routing through a channel network based on the network width function.

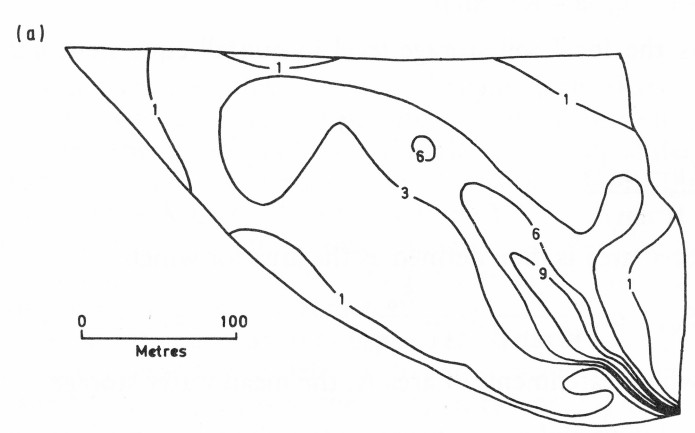

**Figure 1.  Distribution of the topographic index$(a/\tan\beta)$ for the upper East Twin catchment, Mendips, UK (from Kirkby, 1975)**


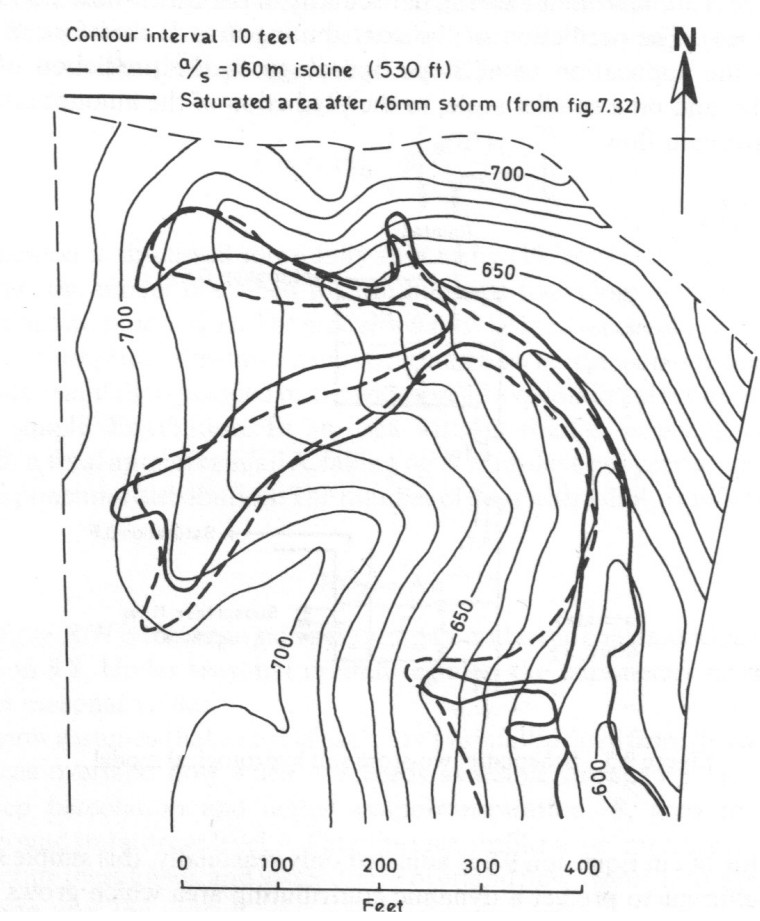

**Figure 2.  Comparison of the topographic index $(a/\tan\beta) > 100$ m²/m with observed saturated areas after spring snowmelt in the Sleeepers River WC-4 catchment, Vermont, USA (from Kirkby, 1978)**

In this form, Topmodel does not require a steady rainfall duration long enough to reach steady state, only that the storage for any given value of $\bar{S}$ is configured *as if* it was at a steady state with a steady homogeneous recharge rate.  This implies that as storage changes the celerities in the saturated zone are fast enough that the transition between configurations with changes in storage are relatively rapid (see Kirkby, 1997).   This will be more likely in wet, relatively shallow soils on moderate slopes, and where soil permeabilities increase with saturation.  Kirkby (1975) then introduced an additional assumption that downslope flow could be represented as an exponential function of storage deficit below saturation, $D$ in units of depth.   It was realised that by expressing the saturated storage in the profile in terms of storage deficit rather than water table depth, one parameter could be eliminated.   At that time, the issue of designing models to facilitate the calibration problem and reduce the potential for overfitting was already the subject of discussion in the literature (Ibbitt and O'Donnell, 1971, 1974; Kirkby 1975; Johnston and Pilgrim, 1976).   Thus



$$q = T_o \tan \beta \exp\left(-D/m\right) \qquad [6]$$

where $T_o$ is the downslope transmissivity when the soil is just saturated, and $m$ is a parameter also with units of depth.   This
can be integrated to give an expression for the catchment recession curve in terms of a mean storage deficit, $\overline{D}$.   Following the
same derivation as above gives the condition for saturation now as

$$ln\left(\frac{a}{\tan \beta}\right) > \frac{\overline{D}}{m} - \lambda \qquad [7]$$


where $\overline{D}$ is the mean storage deficit and $\lambda$ is now the areal integral of $ln(a/\tan \beta)$ with $T_o$ and $m$ assumed spatially constant.
This is the expression used to determine the dynamics of the saturated contributing area in what might be called the classical
version of Topmodel (figure 3).  Equation [6] can be integrated along the length of the channel network to provide the integral
discharge from the hillslopes in terms of the mean storage deficit $\overline{D}$ as


$$Q_b = Q_o exp(-\overline{D}/m) \qquad [8]$$

where $Q_b$ is the integrated output along the channel, $Q_o = Ae^{-\lambda}$ (for the case of a homogeneous downslope transmissivity)
and $A$ is the catchment area (see Beven, 2012, for a full derivation). To initialise the model this relationship [8] can then be
inverted, given a value of catchment discharge to give the initial mean storage deficit.   Note that [8] implies a first order
hyperbolic shape for the recession limb of the hydrograph.   This can be checked for a particular application by plotting the
inverse of observed discharges against time.  This should plot as a straight line if [8], and consequently [6] is valid (Ambroise
et al., 1996a; Beven, 2012).

It is worth noting here that the deficit $D$ here represents a storage deficit due to gravity drainage.  Any additional deficits
resulting from evapotranspiration losses are calculated separately from the various model stores.  By the addition of an
additional parameter of available storage for gravity drainage per unit depth of soil (which can be related to the concept of
"field capacity"), the deficit $D$ can be converted to a depth to the saturated zone.   Thus the model can equally formulated in
terms of water table depths and there are a variety of applications that have used the model in this way to compare against
observed depths of saturation (e.g. Sivapalan et al., 1987; Lamb et al., 1997, 1998; Blazkova et al., 2002; Freer et al., 2004).





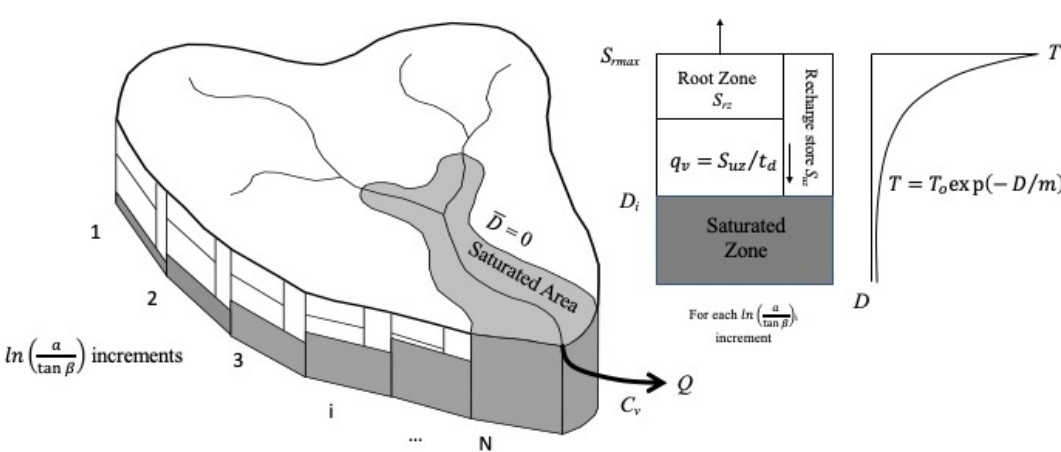

**Figure 3.  Schematic of the classical version of Topmodel (see Table 1 for the definition of the parameters)**

The topographic index acts as an index of hydrological similarity (Beven et al. 1995; Beven, 2012) resulting from the assumption of homogeneous recharge to the satuarated zone at any point in time.  The elegance of the similarity approach means that it is not necessary to make calculations for every point in the catchment only for representative values of the index, which can then be weighted by the distribution function.  This was particularly important when computer power was limited in the 70s and 80s but remains useful in applications to large catchments and when ensembles of runs are required for

uncertainty estimation.  Good resolution is required at the higher end of the distribution where the contributing area first starts to spread, but experience with the model suggested that about 30 representative values was generally sufficient for convergence of the calculated outputs.  Since the pattern of the topographic index is known, one very important feature of the model is that, despite the computational efficiency, the results can be mapped back into space and consequently checked for realism.

We now know that this was not the first analysis of surface saturation of this type.  Horton (1936) came very close to deriving a form of topographic index but restricted his analysis to a single steady state condition, with an input rate equal to the final infiltration capacity of the soil surface (as appears in the Horton infiltration equation).  This, he proposed, suggested a maximum depth of saturation on a hillslope once a steady state at that input rate had been achieved, and could be used to see if the soil would saturate (see Beven, 2004; 2006).  He made no attempt to estimate how long it might take such a steady state

to be reached (but see Beven, 1982; Aryal et al., 2005).  A very similar wetness index was also developed independently by Emmett O'Loughlin (1981, 1986), and was used in the hydrological model of Moore et al. (1988).





Two further components are required to complete the model, to represent the unsaturated zone and routing surface runoff and channel flows. Both showed changes over time. In the BK79 version of Topmodel there were separate interception and

infiltration stores. Evapotranspiration depended on the storage in these stores, with recharge to the saturated zone represented as a constant drainage rate while storage was available. Later these stores were integrated into a single root zone store (to reduce the number of parameters required) and recharge was made more dynamic dependent on the local storage deficit $D$ and storage in the unsaturated zone in excess of "field capacity". This was controlled by a time delay per unit of deficit parameter, $t_d$.


In respect of routing the surface and channel flows, there was one thing that KB got wrong in the original BK79 model formulation. This used a form of explicit nonlinear time delay routing for the overland flow and channel network that will produce kinematic shocks at times when the hydrograph is rising quickly. This was based on the field observations of mean channel velocities derived from a large number of salt dilution gauging experiments that were used to measure overland flow

velocities and check the discharge ratings at the stream gauging sites. Later it was realised that the routing should be based on celerities rather than velocities and that it is possible to have a nonlinear velocity–discharge relationship that produces a constant celerity (e.g Beven, 1979) allowing the use of a stationary time delay histogram in routing the runoff. This, with the advantage of simplicity, was then used in later versions of Topmodel. The resulting set of parameters needed for a model run are defined in Table 1.


**Table 1.  Definitions of parameters in the "classical" version of Topmodel**

| Parameter | Definition |
|---|---|
| $T_o$ | Downslope transmissivity when the soil is just saturated to the surface |
| $m$ | Exponential scaling parameter for the decline of transmissivity with increase in storage deficit $D$ |
| $S_{rmax}$ | Maximum capacity of the root zone (available water capacity to plants) |
| $S_{ro}$ | Initial storage in root zone at the start of a run |
| $t_d$ | Time delay for recharge to the saturated zone per unit of deficit |
| $C_v$ | Channel routing wave velocity (celerity) |



In what follows, some of the history of Topmodel will be recalled. This history will be necessarily incomplete. Topmodel
was always presented as more a set of simple modelling concepts for making use of topographic information in hydrological
prediction than as a fixed model structure (see Beven, 1997, 2012). This has left plenty of scope for others to use those
concepts in different ways or incorporate them into other models. The simplicity and open source distribution of the modelling
code has also resulted in applications, more or less successful in terms of hydrograph fits, many of which have been in areas
where the assumptions should not be expected to be valid. It is also impossible to summarise all those applications that use or
cite Topmodel but a list of the various main developments and uses of the model through time is also provided in the Electronic
Supplement. This history therefore reflects the particular viewpoint of the authors who were involved in the original
development of Topmodel, Distributed Topmodel and Dynamic Topmodel.

## 2. Topmodel: from rejection without being refereed to highly cited


The first Topmodel paper submitted to a journal was rejected without being refereed by the Journal of Hydrology by one of its
editors, Eamonn Nash in a short letter as "being of too local interest" before later being accepted by the IAHS Hydrological
Sciences Bulletin as Beven and Kirkby (1979, BK79 from hereon). This rejection should not be as surprising as it might seem
now, given that this is one of the most highly cited papers in hydrology[2]. In 1978, many computer programs and data were
still stored on cards; even "mainframe" computers had relatively small amounts of memory. Because there were no digital
elevation models, the analysis of catchment topography was a manual and very time-consuming process. The derivation of
the topographic index for the small Crimple Beck catchment where Topmodel was first applied involved the use of maps,
aerial photographs and field work, and took days of intensive work. For an engineer like Eamonn Nash, it was difficult to see
how such an approach could ever be of use to a practicing engineering hydrologist.


On moving to Leeds University, MK obtained a UK Natural Environment Research Council grant to develop the concept into
a computer model of catchment hydrology, with funds to employ a post-doctoral research assistant. The grant also allowed
for running a nested catchment experiment with multiple raingauges and stream gauging sites, together with saturated area
monitoring and other observations. KB was still finishing his PhD work at the University of East Anglia, on a finite element
model of hillslope hydrology, but was fortunate to be appointed to the Leeds post.

Crimple Beck, upstream of an existing River Authority gauging station, was chosen as the field site, and with the help of
technician Dick Iredale, a lot of time was spent instrumenting and maintaining the gauges (both raingauge and water level
recordings at that time were made on charts, and a suite of computer programs was also developed to digitise and analyse the
charts, see Beven and Callen, 1979). Methods were also developed for measuring infiltration rates and overland flow

---

[2] Google Scholar lists >7000 citations in April 2020. Unfortunately, Hydrological Science Bulletin for 1979 and earlier is
not listed on Web of Science; only from when it changed to Hydrological Sciences Journal in 1980.


velocities using a plot sprinkler system (an interesting experience on the windy moors in the headwaters of Crimple Beck, even with a plastic sheeting wind break around the plots). Some of the results of the Crimple Beck process studies, highlighting the differences in response between headwater and sideslope areas are reported in Beven (1978). As a result, the application of the model to the 8 km² Crimple Beck in BK79 made use of different topographic index distributions in 23 headwater and

sideslope subcatchments, each with its own topographic index distribution (figure 4). Each subcatchment could also have a different precipitation input based on interpolation from the network of raingauges that had been installed. At this time, Topmodel went through numerous early versions, initially in hard copy as punched cards, then stored digitally that could be edited using a teletype terminal (a very slow process which required each edit being typed and printed on a roll of paper), and still later with editing on cathode ray tube (CRT) terminals.


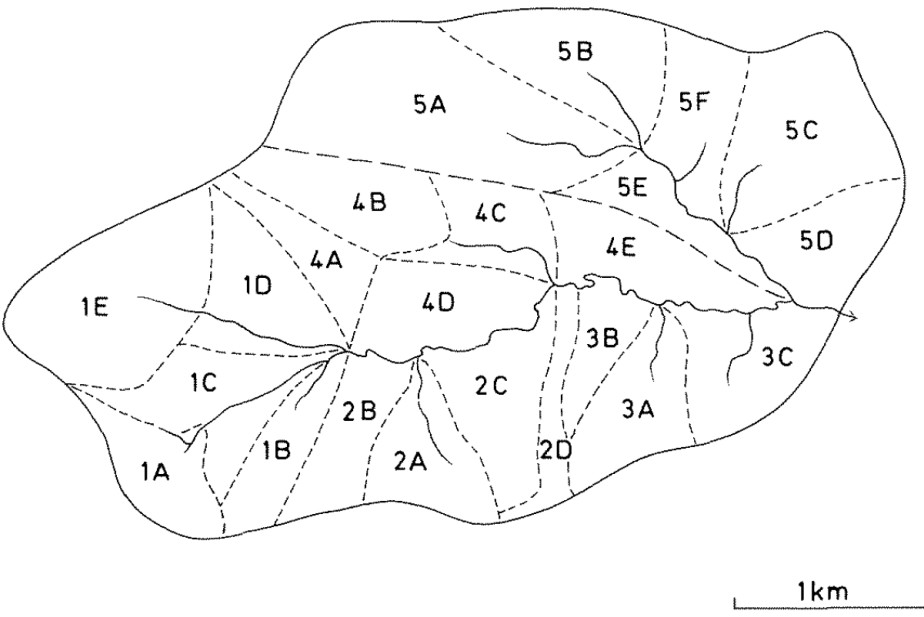

**Figure 4.**   **Subdivision of the 8 km² Crimple Beck catchment, Yorkshire, UK into 23 headwater and sideslope subcatchments (from Beven and Kirkby, 1979).**


One of the aims of the original modelling project was, in fact, to produce a model structure that could be applied on the basis of field measurements alone. The BK79 paper demonstrated how model optimisation produced a parameter set that effectively resulted in the subsurface storage being used as an overland flow store in this wet flashy catchment. Parameters derived from field observations, on the other hand, reproduced the observed saturated areas reasonably well, with the exception of some

sand lenses in the glacial deposits that covered the catchment (see BK79). This work was then extended in the paper of Beven et al. (1984) where, based on the fieldwork of Nick Schofield and Andy Tagg, it was shown that reasonable hydrograph



predictions could be obtained using only field measured parameters. This work is still one of the few papers to demonstrate some success in using parameters derived from field observations, though it is worth noting that the characteristics of the exponential subsurface storage were derived from a recession curve analysis using a limited number of discharge
measurements at the site of interest. This could then be interpreted in the terms of the theory of the model (equation [8]), an approach more appropriate to the scale of application than profile measurements.

## 3. The attractions of Topmodel

The main attractions of Topmodel have always been its elegant simplicity that captures the dynamic and dominant hydrological spatial controls in a semi-distributed form; ease of setting up an initial catchment application; the resulting speed of computation; its ease of modification (it is more a set of concepts rather than a fixed model structure); and its direct link to topography as a control on the hydrological response of a catchment such that predicted storage deficits and saturated contributing areas can be mapped back into space. Whilst its simplicity has a firm theoretical basis. However, the simplicity
comes at the cost of assumptions that mean that the model might not be applicable everywhere but were always clearly identified. Early in the days of digital elevation models (DEMs), topographic index values were calculated for the whole of the conterminous US (see more recently the global study ofa Marthews et al., 2015, using the HydroSHEDS database). The data were available to do so using a digital terrain analysis, but no hydrologist should expect that the basic Topmodel concepts would be suitable for the whole of the conterminous United States (nor for many other areas of the world that are flat, or with
deep subsurface flow systems). It might be possible to calibrate a version of the Topmodel to give hydrograph predictions for such catchments, but that does not mean that the assumptions are valid, or that the mapping of storage deficits back into the space of the catchments will be meaningful.

This is indicative, however, of why Topmodel has proven so popular and highly cited over the years. Topography is in general
important to the flow of water in hillslopes. As soon as digital elevation models started to become more widely available in the 1980s onwards, hydrological modellers have wanted to make use of them in some way. But given that information about topography, what to do with it? All the time that a time consuming manual analysis was required, Eamonn Nash was right; other more conceptual modelling approaches were more attractive. But given the possibility of a DEM and a digital analysis, suddenly ways of using topography in modelling became much more attractive., especially given the available software for
digital terrain analysis and other geographical information overlays. Effectively, once the topographic index distribution had been calculated, like many other conceptual hydrological models only input precipitation and potential evapotranspiration time series were needed to make a run (and the latter was even made available as an option within Topmodel as a simple parameterised sinusoidal function following the work of Calder et al., 1983, for use when other estimates were lacking).



Various model structures can make use of either gridded or triangular irregular network topographic data, but of those available Topmodel provides the simplest and fastest approach.   It has been including in a variety of general hydrological modelling packages including FUSE (Clark et al., 2008), SuperFLEX (Fenicia et al., 2011); and MARRMoT (Knoben et al., 2019), though none of these provide facilities to compute the topographic index.   In the 1980s and 1990s, the storage and analysis of large DEMs was still a computationally significant problem.   Topmodel required that an analysis could be carried out just

once prior to running the hydrological model, after which only the statistical distribution of the topographic index was required to run the model but, if required, the results could still be shown as maps because of the explicit link between location and the topographic index.

## 4.   The Early Days of Digital Terrain Analysis


 There remain, however, issues about how to process the DEM to determine slope and upslope area, particularly for square gridded data.   The application of Topmodel requires an analysis of terrain data to define the pattern of the topographic index. It has already been noted that in the original application to Crimple Beck, this was an extremely time-consuming process.   It involved working with maps and air photographs to determine the apparent flowlines and hillslope segments, then calculating

slopes between contour lines and areas with a planimeter.   This had some advantages in that features such as gullies and ditches that could be observed in the field or in air photographs could be taken into account.   It involved some decisions about what to do with the small, often triangular, sections that were left where contours crossed a river (figure 5).   An alternative approach was suggested by Beven and Wood (1983) of representing various hillslope elements making up the catchment as geometric forms of varying width and slope, from which the topographic index could be derived analytically.


Later (around 1976), this process was partly computerised by noting the coordinates of intersections between flowlines and contours and typing them onto punched cards (on an IBM029 card punch) that could be input and processed by computer. Later still (around 1978), KB had moved to the Institute of Hydrology at Wallingford where early work on digital terrain analysis was being carried out, including the digitising of contour maps on a large digitiser.   KB made use of this to speed up

the process of inputting the data for processing.   It was not until 1982, when KB returned to the Institute of Hydrology from working at the University of Virginia, that there was access to gridded digital elevation data.

In fact, KB already had some experience of working with digital elevation maps having carried out an undergraduate project at the University of Bristol on determining flow networks on randomly generated triangular elevation grids with the aim of

looking at the variability in Horton's Laws (following Shreve, 1967). This is relatively simple on a triangular grid, but t more assumptions are needed for a square grid.   This was the start of work on the multiple downslope direction flow algorithm (now often called the MD8 algorithm) that was later published in the Topmodel application of Quinn et al. (1991, also Quinn et al., 1995a) and independently by Freeman (1991).      The Topmodel digital terrain analysis software for gridded data



(DTMAnalysis) was made freely available in the 1990s as a Visual Basic program, including sink filling, catchment
delineation, and topographic index derivations (e.g. figure 6).   Other DEM routing algorithms have also been used, see for
example Wolock and McCabe (1995), Tarboton (1997) and Pan et al. (2004).

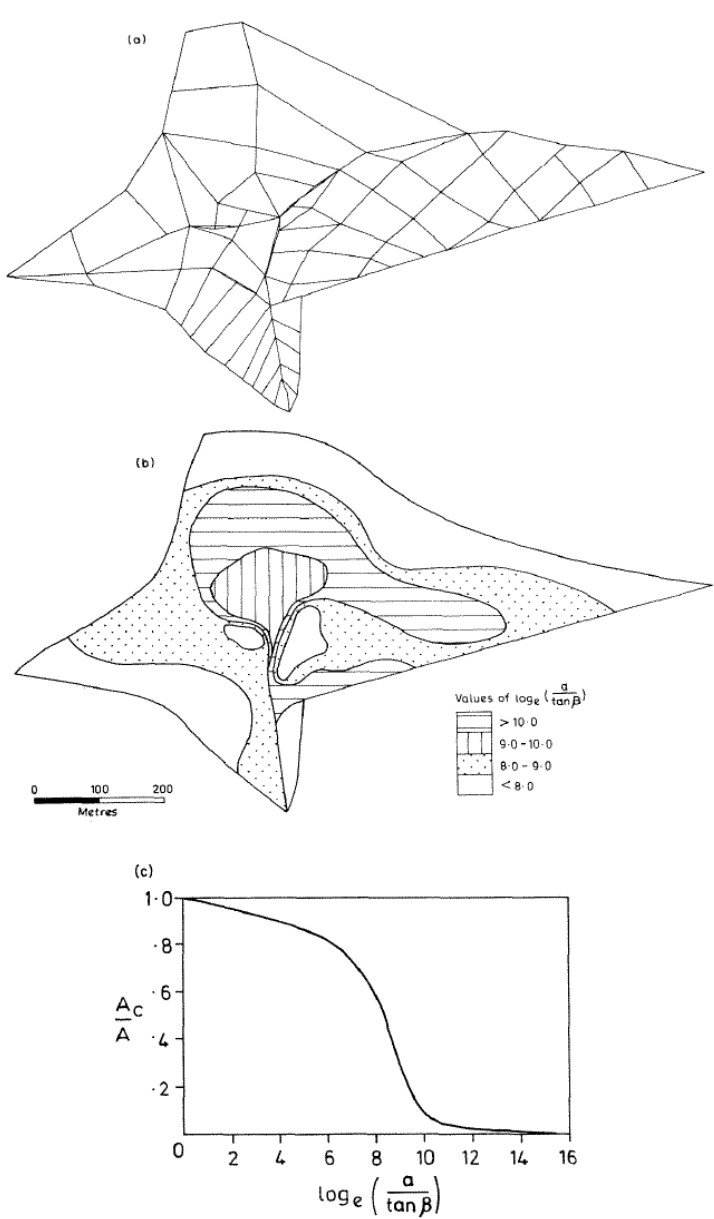

**Figure 5.   Manual topographic analysis of the Lanshaw subcatchment of Crimple Beck, showing the discretisation**
295              **and pattern and distribution function of the topographic index (from Beven and Kirkby, 1979)**


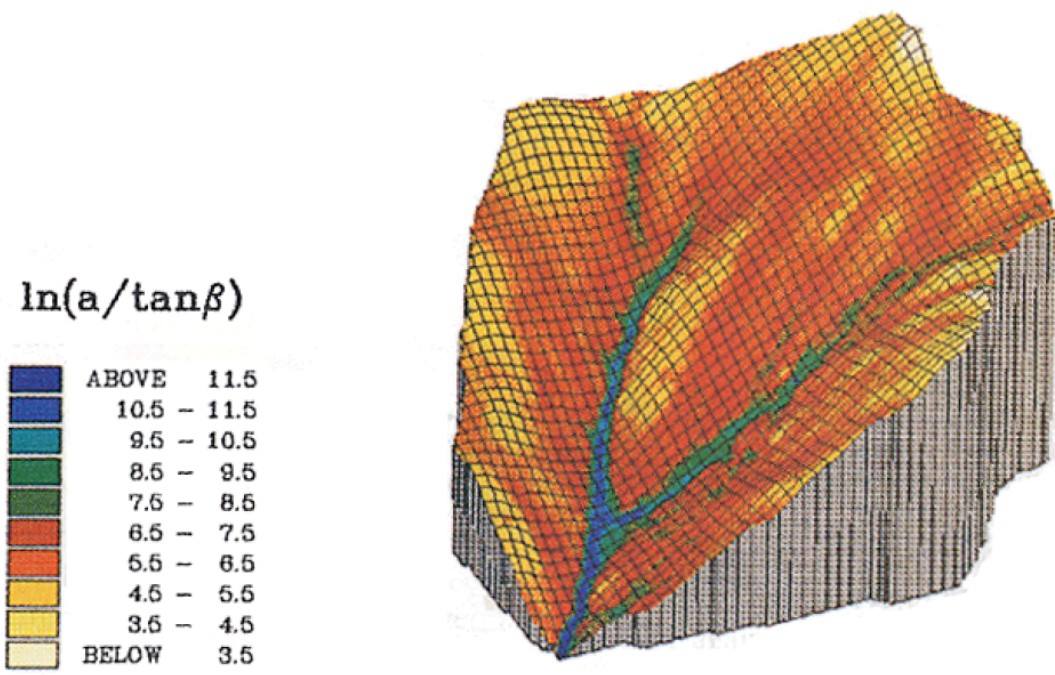

**Figure 6. Pattern of topographic index for the Rigelbach catchment, Vosges, France superimposed on a digital terrain model (after Ambroise et al., 1996b). The highest values in the valley bottom and convergent hollows will be**

**predicted as saturating first. A small spring in the catchment on the right-hand hillslope, indicating subsurface convergence, is not reflected in the pattern of the index shown on this map since this is based on the topographic flow pathways alone.**

There were also other aspects to the early days of digital terrain analysis, in particular that the early (relatively coarse) gridded data sets were not necessarily hydrologically consistent i.e. the mapped blue line river network did not always match the lowest points in the digital data. There were also many sinks without outlets, apparently discontinuous rivers, and depending on how the data were processed catchment areas that were incomplete with respect to the contours, or that had gained area from adjacent catchments. All of these issues required either manual intervention or assumptions about how to process the data

(e.g. do you raise sinks until there is a downslope pixel or burrow through a barrier to a lower downslope pixel). The Institute of Hydrology was instrumental in developing a more hydrologically consistent 50m digital elevation map for the UK in the 1980s (see, for example, Morris and Heerdegen, 1988).

## 5. Evaluating the Topmodel assumptions






The simplicity of Topmodel has also been criticised (not least by Beven, 1997, and Kirkby, 1997). In particular the three main simplifying assumptions on which the model is based all have been criticised. As stated in Beven (2012, p210) these are:

A1 There is a saturated zone that takes up a configuration as if it was in equilibrium with a steady recharge rate over an upslope contributing area $a$ equivalent to the local subsurface discharge at that point.

       A2 The water table is near to parallel to the surface such that the effective hydraulic gradient is equal to the local surface slope, $s$


       A3 The transmissivity profile may be described by an exponential function of storage deficit, with a value of $T_o$ when the soil is just saturated to the surface (zero deficit).

Some support for assumption A1 has been given by Moore and Thompson (1996) for a catchment in British Columbia, though
their sample of water tables were mostly near to the stream and measured infrequently and they suggest more work to assess the limits of validity of the assumption. The assumption has been criticized by Barling et al. (1994) and others who noted that the effective upslope contributing area ($a$ in the topographic index) will, in many catchments, be variable as the catchment wets and dries. This was also demonstrated by Western et al. (1999) in the Tarrawarra catchment, where observations of topsoil water content showed that topography can be a control on soil water content in wet conditions but that the pattern will
be much more random in dry conditions, reflecting evapotranspiration rather than topographic controls on the patterns of moisture. This should not be a surprise at Tarrawarra which has duplex soils with a shallow active layer underlain by an impermeable subsoil. In dry conditions, evapotranspiration will dominate the pattern of soil moisture in the topsoil; Topmodel has a root zone storage to deal with this quite separate from the treatment of downslope flows. It is clear, however, that the potential for a dynamic $a$ will be an issue in many catchments, at least seasonally in the transitions from wet to dry conditions.
Seibert et al. (2003) also showed that in the Svartberget catchment in Sweden, there was a high correlation between water table levels and distance to the nearest stream, even in upslope areas, but that the patterns over time suggested that assumption A1 was not valid there.

Modifications to Topmodel have been suggested to allow a dynamic recalculation of the topographic index distribution under
wetting and drying (Barling et al., 1994; Piñol et al., 1997; Saulnier and Datin, 2004; Loritz et al., 2018) and there is an increasing appreciation that connectivity of both surface and subsurface flows on hillslopes is one reason for the nonlinearity of hydrograph responses and the threshold behavior of runoff generation in small catchments (see for example, Graham et al., 2010). The A1 assumption implies that there is always connectivity, while in the original TOPMODEL any overland flow



generated on a topographic index increment is assumed to reach the stream. In some situations this will not be unreasonable
in that if an area generates fast runoff frequently (on areas of low slope, or areas of high convergence) there will often be a rill
or small channel that conveys that runoff downslope, even if that area might be some way from a channel. In other areas, run-
on effects will be important in increasing soil water content and saturation downslope. Where this is important it can be
represented in the Distributed version of Topmodel, and Dynamic Topmodel (see below). However, those small rills and
channels are usually too small to be seen in even fine resolution DTMs, so might be missed in setting up a more detailed model.
They can sometimes be clearly seen in the field or from aerial photographs as having different, wetter, vegetation patterns.

Another criticism of A1 has been that the water table on a hillslope is never in steady state. But as noted earlier that is not
exactly what A1 actually says. The starting point is the saturated zone storage (determined by water balance accounting); A1
says only that the configuration of the water table will be _as if_ that storage was in steady state given the discharge at that time
step. Clearly this is an approximation but Kirkby (1997), for example, shows how the quasi-steady state assumption that
underlies the topographic index can be justified by a kinematic wave analysis of celerities on a hillslope, particularly for the
exponential transmissivity profile case.

A2 has also been criticized when there are deeper flow pathways. Groundwater analyses suggest that deeper water tables will
not be parallel to the surface and may even involve upward fluxes and cross-divide fluxes between catchments. Where this is
important then clearly the Topmodel assumptions will not be valid, but the assumption can be relaxed. Quinn et al. (1991),
for example, showed how the topographic index can be derived using a reference slope pattern for the water table rather than
the surface slope. Use of Topmodel in this context then assumes that the water table is always parallel to the reference pattern
(except where it intercepts the surface). This will also be an approximation but allows the Topmodel concepts to be applied
to a wider range of situations (it might also require use of a non-exponential transmissivity profile, and to allow for the different
depths of unsaturated zone that might lie above points with similar reference level topographic index values).

Beven (1982a, 1984) showed that the exponential assumption of A3 could be justified for at least some soils (see also Michel
et al. 2003), although Franchini et al. (1996) show how calibrated values of the surface transmissivity values tend to be high,
and linked to the grid scale used in the topographic analysis. This dependence was investigated further by Saulnier et al.
(1997a,b), Ibbitt and Woods (2004); and Ducharne (2009) who suggested ways of correcting for it, and by Pradhan et al.
(2006, 2008) who used fractal scaling arguments to adjust topographic index distributions from coarse to fine scales to stabilize
parameter estimates.

One criticism of A3 has that the recession limb of hydrographs is not always of the first order hyperbolic function of time that
an exponential transmissivity function implies. That is not too great a problem in that, as noted above, different types of
transmissivity profile representing different shapes of recession can be assumed, but which imply a change in the definition of





the associated topographic index (Ambroise et al., 1996a). Other groups have taken a different approach by modifying the Topmodel concepts to allow for more complex process representations in different catchments. In particular, additional

storage elements have been added to simulate shallow subsurface stormflows when the exponential store of the original Topmodel did not appear to hold (e.g. Scanlon et al., 2000; Walter et al., 2002; Huang et al., 2009).

A further criticism has been that A3 does not properly account for the transient downslope flows in the unsaturated zone. We can conclude that it is valuable to evaluate the perceptual model of the characteristics and processes in a catchment to decide

which sets of assumptions might be more plausible (see, for example, Piñol et al., 1997; Gallart et al., 2007; Beven and Chappell, 2020).

## 6. Extensions to the classic Topmodel concepts

In addition to the extensions and relaxations to the original model formulation discussed in the previous section, Beven (1982b) proposed an extension to the theory to allow for heterogeneity in the soil profile characteristics in a catchment by use of a soil-topographic index $ln(a/T_o \tan \beta)$ (see also Beven, 1986, 1987) . If it is assumed that the soil is everywhere homogeneous then $T_o$ will have no effect on the spatial and cumulative distribution of the index; but if there is evidence to allow it to vary within the catchment then the variability in $T_o$ will change both the pattern and cumulative distribution of the saturated

contributing areas. If soil depths vary, this might also require allowing for different depths to the saturated zone for similar values of the index (Quinn et al., 1991; Saulnier et al., 1997c). The soil-topographic index was used in two studies in catchments where many piezometers were available to indicate patterns of saturation (Lamb et al. 1998; Blazkova et al., 2002), allowing local transmissivities to be defined. Interestingly, in both cases, this resulted in a steepening of the cumulative distribution of the index suggesting a later onset of a saturated contributing area but a more rapid spread once it was established.

Greater heterogeneity in soil permeability also means that there is a greater potential for infiltration excess overland flow and Beven (1984) provided a Green-Ampt type solution for infiltration capacity that was consistent with an exponential hydraulic conductivity assumption (see also Larsen et al.,1994). This was implemented in some versions of Topmodel assuming isotropy of vertical and downslope conductivities.

Further extensions were proposed for cases where the catchment recession is not consistent with the exponential storage/flow function of BK79. This was extended to other forms of storage-discharge relationship by Ambroise et al. (1996a) (Figure 7), Iorgulescu and Musy, (1997), and Duan and Miller (1997). These forms then imply the use of a different form of topographic index to $ln(a/\tan \beta)$, and might also preclude the use of the implicit redistribution of subsurface storage (see Kirkby 1997). A generalised formulation for an arbitrary empirical recession curve was also proposed by Lamb and Beven (1997) and Lamb

et al. (1998).





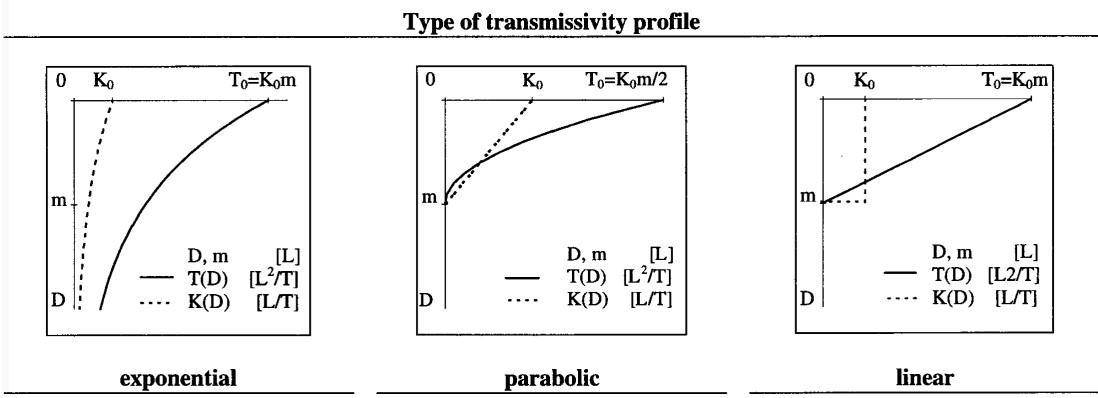

**Figure 7.   The different types of transmissivity profile considered in Ambroise et al. (1996a).**


## 7.   Evaluating the spatial predictions of Topmodel

As noted earlier, one of the most important features of Topmodel is the possibility of assessing the spatial pattern of predictions of storage deficits, saturated areas, or water tables.   The earliest evaluations of the spatial predictions of Topmodel were in

Kirkby (1978, Figure 2) and in the original BK79 paper.   This was based on field work in the small Lanshaw headwater subcatchment of the Crimple Beck evaluation where a network of over 100 overland flow detectors was installed.   These were simple T-tubes of plastic pipe, with holes in the top of the T at ground level such that water would collect in the vertical tube if overland flow occurred.   This is a very simple technique but, of course, only gives a binary measure of occurrence and requires visiting the network (and being able to find all the tubes) after every storm.   This was a significant effort but allow

percentage saturation statistics to be built up over a number of storms.   This showed that saturation in this subcatchment was related to storm peak discharge but peaked at about 95%, whereas the model would predict up to 100% saturated contributing area.   Further investigation showed that this difference was due to two areas of more permeable fluvioglacial sand in the catchment that were much less likely to saturate.   Even this small catchment was not homogeneous in its soil characteristics.

This is one of the issues in doing this type of comparison (and of setting up hydrological models anywhere since without such local knowledge they cannot be right in detail).   Two types of state observations are generally used in model calibration or evaluation: percentages or maps of saturated areas at one or more time steps; and point measurements of water tables.   Another evaluation of mapped saturated areas at the scale of a small catchment was carried out by Franks et al. (1998) in the *bocage* landscape of Brittany.   They investigated the potential of airborne radar to detect valley bottom saturated areas.   This turned

out to be limited by the difficulty of distinguishing saturated from near-saturated areas but, in that landscape, the wet areas corresponded closely to areas traditionally walled off to keep the cattle out, something that could be used over wider areas in model evaluation.



At a larger catchment scale (40 km²) Güntner et al. (1999) mapped out saturated areas by field surveys in the Brugga catchment
in Germany using pedological and vegetation characteristics, comparing the results with the Topmodel predictions (for a single
optimised parameter set). Their conclusions are an indication of the type of match that might be achieved at this scale: "*Their
[saturated areas] mean simulated percentage on total catchment area was about 5.5% (Table III) which corresponded well to
the mapped percentage of 6.2%. On the other hand, the simulated percentage of saturated areas was highly variable with time
(Figure 5 and Table III). During high flow periods it reached nearly 20%. This was in contrast to the field observations, where
spatial variability of the extension of saturated areas was small. A percentage higher than 10% was not reasonable in the
study area, except for extreme situations, which did not occur during the study period. In the model, because of the large
percentage of simulated saturated areas during floods, overland flow rates and consequently total runoff. would be simulated
too high. For compensation, parameter m had to be calibrated to a large value in order to better match observed peak flow at
the expense of the performance of recession simulation. This is due to the function of this parameter to control the dynamics
of subsurface runoff, with lower m reducing the range of subsurface flow rates and, thus, diminishing peak flow but also
flattening out recessions. In summary, the poor correspondence of calibrated m to its value derived from the recession analysis
revealed that the calibration of m was influenced by inadequacies of the model structure for the study area, i.e. an
overestimation of the dynamics of saturated areas.*" (pp 1616/1617)

A number of studies have compared Topmodel spatial predictions to observed patterns of water tables and mapped saturated
areas with more or less success (e.g. Ambroise et al., 1996b; Moore and Thompson, 1996; Seibert et al., 1997; Lamb et al.,
1997, 1998; Blazkova et al., 2002; Freer et al., 2004). Two issues arise in comparing observed and predicted water tables.
The first is converting predicted modelled (gravity drainage) storage deficits to water table depths; the second is the
commensurability issue of comparing the modelled variable, representing some average over a topographic index increment
to local point observed values. These may be given the same names by the hydrologist (soil moisture, water table depth,…)
but represent different quantities when they reflect different scales (see, for example, Freer et al., 2004 who allowed for sub-
grid uncertainty in the model evaluation). This is a particular problem when no information is available about the spatial
variability of transmissivity in the catchment, so that it is necessary to assume a homogeneous transmissivity in the model.
These issues mean that even if the Topmodel assumptions might be a reasonable simplification in modelling a catchment, we
would not expect the predictions to match the observations exactly (Lamb et al., 1998; Blazkova et al. 2004). Defining
saturated areas relative to the grid scale of the topography and topographic index can also be an issue (Gallart et al., 2008). It
also means that the match can be improved by the back-calculation of a local transmissivity at each observation point or
mapped saturated area boundary to give better fits to stream discharges, though point observations did not prove to have the
effect of also reducing the uncertainty in predicted discharges (Ambroise et al., 1996b; Lamb et al. 1998; Blazkova et al.,
475    2002).


There is also the possibility that subsurface flow lines might not follow the surface topography producing concentrations of saturation, for example, as the result of fracture systems in the bedrock. This has been found in the Ringelbach catchment (Ambroise et al., 1996b) and the Slapton Wood catchment (Fisher and Beven, 1996) but of course is very difficult to
incorporate in any model without a detailed characterisation of the subsurface. Freer et al. (1997, 2002) found that at the Maimai and Panola catchments better characterisation of the water tables was achieved using a topographic index based on the bedrock topography (defined at great effort on a 2m grid with a knocking pole) rather than the surface topography. This was related to collection of flow in hillslope trenches (although a significant amount of flow was also collected from discrete macropores in the soil). Obtaining such information over larger areas is, however, much more difficult, even using geophysical
methods, and often there is not such a clearly defined transition to bedrock.

It is then interesting to consider how good the spatial predictions should be before the Topmodel assumptions are considered invalid. If we look in enough detail, all model hypotheses have their limitations, but in making an evaluation it is also necessary to consider the uncertainties in the forcing and evaluation data and the commensurability issues of comparing
observed and predicted variables (see the discussion of Beven, 2019). Blazkova et al. (2002) considered whether the death of Topmodel should be declared on the basis of evaluations of both hydrograph and water table predictions and suggested that, at least for the catchment studied, such an announcement might still be premature. But, we repeat, the Topmodel assumptions will apply to only a subset of catchments, and perhaps to only a subset of catchments for which applications of Topmodel have previously been published. One of the reasons for the development of the Dynamic version of Topmodel (see below) was to
relax some of the spatial homogeneity assumptions of the original model.

As noted previously, one of the features of the Topmodel formulation is that the topographic index on which it is based, both has a physical basis as an index of similarity and allows a computationally efficient code. It may not, however, be the best index of similarity in all catchments, and there have been a number attempts to formulate alternative forms. In particular,
indexes based on height above the nearest river channel (Crave and Gascuel-Odoux, 1997; Rennó et al. 2008; Gharari et al., 2011), and an extension of this based on consideration of the dissipation of potential energy (Loritz et al., 2019) have been proposed and tested in discriminating different hydrological responses within catchment. Other approaches have included the travel time index of Barling et al. (1994), the variable recharge index of Woods et al. (1997); the downslope wetness index of Hjerdt et al., (2004), and the hillslope Peclet number of Berne et al. (2005). Interestingly, unlike the Kirkby index used in
BK79 or the O'Loughlin wetness index, most of these do not explicitly consider the effects of hillslope convergence or divergence on saturation and runoff processes. There may be an implicit effect, in that areas of convergence near the base of hillslopes will have a greater area with little elevation difference to the nearest stream, relative to divergent slopes that are more convex in form.

**8.  Topmodel calibration and uncertainty estimation**





One of the original aims of the development of Topmodel was to keep the model structure simple and as parametrically parsimonious as possible while still retaining the possibility of mapping the model predictions back into space and determining the model parameters by field measurement, as in Beven et al., (1984). Table 1 presents the parameters that need to be defined

in the classic version of the model. The 1970s was a period when most model applications involved manual calibration, although there had been significant research on the application of automatic computer calibration methods to hydrological models. Automatic methods were still somewhat limited by the computer resources available, especially for models that had large numbers of parameters or were slow to run. Norman Crawford who, as the PhD student of Ray K. Linsley, developed the Stanford Watershed Model (that later developed into the HSPF package) argued that manual calibration was advantageous

in that hydrological reasoning could be used in the calibration process. The Stanford model, however, had many more parameters than Topmodel and there was a rumour at the time at the only person who could successfully calibrate the Stanford Model in this way was Norman Crawford (Crawford and Linsley later founded the Hydrocomp consultancy company to promote the Stanford Model, see Crawford and Burges, 2004).

In fact, the original BK79 Topmodel paper includes a comparison of field measured and optimised calibrations (as determined from response surface plots) to the Lanshaw subcatchment of the Crimple Beck. This proved to be interesting in that the optimisation produced slightly better goodness-of-fit but took the model into a part of the parameter space that meant that contributing area component was entirely eliminated and the whole basin response was simple being represented by the exponential store. This was not perhaps surprising in this relatively wet, rapidly responding catchment but the manual

calibration was able to ensure that the model functioned as intended (consistent with the perceptual model on which it was based). KB was always very wary of optimisation methods for model calibration as a result of this experience.

This has not prevented the use of automatic optimisation by others, however. Topmodel was quick to run (once the topographic index distribution had been determined) and so well suited to automatic methods. That also meant that it was

also well suited to the use of random parameter sampling or Monte Carlo methods. KB made the first Monte Carlo experiments with Topmodel in 1980 when working at the University of Virginia (UVa) in Charlottesville with access to a fast (for its time) CDC6600 "mainframe" computer. This work was inspired by the Regionalised or Generalised Sensitivity Analysis (GSA) methods developed by George Hornberger (also at UVa), Bob Spear and Peter Young (see Hornberger and Spear, 1981). The GSA approach differentiated between sets of "behavioural" model parameters and those considered "non-behavioural". KB

extended this binary classification to express some of the uncertainty associated with the model predictions, by weighting the outputs from each model run by an informal "likelihood" based on a goodness-of-fit measure. Non-behavioural sets of parameters are given a likelihood of zero and do not contribute to the prediction uncertainty.





This was the origin of the Generalised Likelihood Uncertainty Estimation (GLUE) methodology that was first published more than a decade later in Beven and Binley (1992). The use of informal likelihoods in GLUE proved to be rather controversial relative to statistical methods (see Beven et al., 2008; Beven and Binley, 2014), but the methodology has been used extensively, including in applications of Topmodel and Dynamic Topmodel (as well as with many other models). GLUE does not require a formal statistical model of the residual errors which can be difficult to specify for dynamic models subject to epistemic uncertainties (see Beven, 2016). The first published application of GLUE to Topmodel, appears to have been that of Beven

(1993), closely followed by Romanowicz et al. (1994) (which did use a formal statistical likelihood within the GLUE framework with resulting overconditioning), and Freer et al. (1996), who showed how the distributions of model residuals could be non-Gaussian and non-stationary, and how the likelihood weights could be updated as more data became available.

There have been many other applications of Topmodel and Dynamic Topmodel within the GLUE framework that have
included the use of internal state data in model evaluation as well as discharge observations (e.g. Ambroise et al., 1996b; Lamb et al., 1998; Freer et al., 2004; Gallart et al., 2007) which, it would be hoped, would help judge whether a model is getting a reasonable fit to the data for the right reasons (Klemeš, 1986; Beven, 1997; Kirchner, 2006). It has also been shown how, even in a catchment where the Topmodel assumptions might be considered to be reasonable, some seasonal variation in plausible parameter sets could be identified on the basis of non-overlapping distributions of behavioural parameter sets for
sub-annual periods. Freer et al. (2003) and Choi and Beven (2007) showed how such variation could be incorporated into making predictions by defining classes of hydrologically similar periods; but in both studies this is also an indication of the limitations of the simple Topmodel structure which could, in this case, have been rejected. A similar period classification approach to calibration has been taken more recently by Lan et al. (2018)

Most recently, rather than using an informal likelihood, GLUE has been applied using limits of acceptability that are specified based on what is known about uncertainties in the input and evaluation data before making any runs of the model (Liu et al. 2009; Blazkova and Beven, 2009a; Coxon et al., 2014). This is similar to earlier applications of GLUE based on fuzzy measures and possibilities (e.g. Franks et al., 1998; Freer et al., 2004; Page et al., 2007; Pappenberger et al., 2007). This acts as a form of hypothesis test in conditioning the model space to those areas where plausible models are consistent with the
limits of acceptability. It also allows that all the models tried might be rejected (e.g. Hollaway et al., 2018), although in doing so care must be taken to properly assess uncertainty in the available data (Beven, 2019).

**9.   Topmodel and flood frequency estimation**





One important type of application of Topmodel was to make use of its simplicity and computational efficiency to extend the prediction of hydrographs to flood frequency analysis. The first applications to frequency analysis were part of the PhD thesis of Murugesu Sivapalan at Princeton University. This built on the seminal derived distribution approach of Eagleson (1972),

using a distribution of storm events to drive the model on a storm by storm basis to generate the distribution of flood peaks, including both infiltration excess and saturation excess runoff generation as a function of the topographic index and antecedent wetness. Sivapalan et al. (1987, 1990) showed how the flood frequency distribution could be expressed as a function of non-dimensionalised rainfall and Topmodel parameters. An important simplification in this work was the neglect of any redistribution of subsurface storage within each event; saturation would occur only by volume filling given the contributing

area and deficit at the start of the event. Another aspect of this work was the introduction of the concept of the "Representative Elementary Area" where a similar version of Topmodel was used, with rainfalls assumed to statistically homogeneous with a certain correlation length, to assess the scale at which *pattern* in hydrological heterogeneity became less important, although the nonlinearities arising from the *distribution* of that heterogeneity might still be important (Wood et al., 1988).

The need to generate antecedent conditions for each storm could be avoided by using the model in continuous simulation mode. This allows the antecedent conditions for each event to be consistent with the sequence of previous events. This became possible as more computer power became available, allowing very long runs with hourly time steps to assess the frequency statistics of rare events. This had been done before using of long observed rainfall sequences driving a variety of hydrological models (e.g. Thomas, 1982; Calver and Lamb, 1995), but Beven (1986, 1987) first combined stochastic rainfall

and evapotranspiration generation with Topmodel on an hourly time step as a way of producing sequences of flood peaks for the Plynlimon catchments in Wales. Other applications included catchments in the Czech Republic (Blazkova and Beven, 1997).

More computer power still, notably with the use of parallel PC clusters, allowed this approach to be applied with sets of

behavioural model parameters determined by comparison with historical discharges at a site within the GLUE framework (e.g. Cameron et al., 1999, 2000a,b; Blazkova and Beven, 2002). One of the results of such calibrations was that the annual maximum frequency distribution could be matched quite well, but not necessarily with the same storms in each year of record, due to the uncertainty in both model predictions and observed discharges. A second result was that in doing the comparison it was important to compare like with like. Particularly in smaller catchments the instantaneous flood peak frequency

distribution could have a quite different form to the distribution of peaks for longer time steps (as predicted by the model). A third issue is that when the rainfall model or hydrological model are calibrated against a relatively short period of record, that period may not be representative of the longer term frequency characteristics (e.g. Cameron et al., 2009c; Blazkova and Beven, 2009a).





This work has included some very long runs of the model (multiple sequences of 100000 years) in order to assess the frequency of extreme events for dam safety assessment (e.g. Blazkova and Beven, 2004).   In doing so there are issues about the stochastic generation of very extreme rainfall events.   Where the underlying distributions in the stochastic input generator are assumed to have infinite tails some physically unrealistic storm volumes can be generated.   This can be avoided either by using a modified distribution (e.g. Cameron et al., 2000c) or by limiting storms to a local estimate of probable maximum precipitation

(Blazkova and Beven, 2004).   The latter, of course, can also be controversial but is often used in dam safety assessments.   One advantage of the continuous simulation approach to dam safety is that both the magnitude of the flood peak, and the total volume of runoff supplied can be assessed.   The biggest threat will not necessarily come from the storm with the highest peak if that peak is of short duration (Blazkova and Beven, 2009b).

Another important question is how the frequency of floods might change with changes in climate.   The continuous simulation approach using Topmodel has been applied in this context, including taking account of the uncertainty in reproducing past hydrograph data (e.g. Beven and Blazkova, 1999; Cameron et al., 2000b).   Any such estimates can only represent potential scenarios because of the dependence on estimates of changes in precipitation and other weather variables provided by the climate models.   They thus cannot be associated with any reliable estimates of probabilities such that there may be better ways

of being precautionary about future changes (Beven, 2011).   This work did produce one interesting insight, however.   In general the change in the mean estimate of a rare event (say with 0.01 annual exceedance probability) was much less than the uncertainty of estimating that event under current conditions.   However, the steepness of the cumulative distribution function for such an event could imply a significant change in the risk.


## 10.   A Distributed Topmodel

It is not necessary, of course, to group pixels together into classes based on the topographic index.   The model could equally be run as a fully distributed model.   The obvious advantage of doing so is that the routing of surface runoff can be more

explicitly linked to the topography, and more flexibility is possible in defining pixel characteristics.   Gao et al. (2015) have followed this route, combining the implicit subsurface redistribution of storage based on assumptions A1 and A2 with a stochastic surface flow routing algorithm.   This is based on a mean velocity linked to surface storage in a pixel, an exponential distribution of velocities for surface runoff parcels of water that results in different travel distances for the parcels, and a probabilistic weighting of directions based on the local downslope topography.   This results in more diffuse patterns of runoff

in both space and time, but at the expense of significantly more computational expense.   Gao et al. (2016, 2017) show how the flexibility of the distributed form of the model can be used to represent land management patterns and changes in upland peatland catchments in the UK in this way.





## 11. Developing Dynamic Topmodel

A further step is then to make the subsurface routing also more dynamic by relaxing assumption A1 to create a more dynamic model both in the subsurface storage-discharge relationship and in the treatment of the effective upslope area. The obvious staring point, for the type of shallow, humid, sloping systems for which the classical Topmodel was intended is to formulate the model within the framework of a kinematic wave equation. Kirkby (1997) did this for a single hillslope segment, noting that a more dynamic subsurface routing would particularly be required for transmissivity functions other than the original exponential form. Beven and Freer (2001) later created Dynamic Topmodel which uses kinematic wave routing for subsurface flows between classes of "hydrologically similar" points in a catchment, where the classification need not be based on a form of (a/tanβ) index alone. This then requires a digital terrain analysis that keeps track of all the pathways between one similarity group and others, including discharges to the river network, If does, however, allow much more flexibility in allowing spatial patterns of catchment characteristics into account to reflect an appropriate perceptual model of catchment responses, in that different similarity classes can have different structures and parameters. However, this flexibility is at at the cost of introducing more parameters to be defined or calibrated which is not generally simple to do. The root zone and evapotranspiration components of Dynamic Topmodel were carried over from the original version, but also have been made more complex elsewhere (e.g. in the HYDROBLOCKS code of Chaney et al., 2016). Dynamic Topmodel has been applied in a number of studies including the Panola (Peters et al., 2003), Plynlimon (Page et al., 2007), Maimai (Beven and Freer, 2001; Freer et al., 2004), Attert (Liu et al., 2009) and Brompton (Metcalfe et al., 2017) catchments.

A somewhat similar approach, but applied on a gridded DEM as a fully distributed model, has been taken in the DVSHM model of Wigmosta et al. (1994) and Adriance et al. (2018). Care then needs to be taken in the numerical implementation, particularly in the gridded approach, as kinematic shocks can arise where there are changes of slope or asymmetric convergent hollows. This can lead to numerical dispersion or instabilities, particularly in an explicit time stepping solution. Instabilities can be avoided by applying a 4-point kinematic wave solution at a pixel level, where all upslope inputs have already been solved and can be added so that only the downslope flow is unknown (Beven, 2012). The original Dynamic Topmodel code was written in Fortran 77 and has been later modified into the DECIPHeR Fortran 2008 code of Coxon et al. (2019) which involved a number of important changes to explore simulations over national scale domains, simulating hundreds of catchments. A version in R was provided by Metcalfe et al. (2015) including tests of the effects of spatial and temporal resolution. Metcalfe et al. showed that convergence of the hydrograph predictions require a discretisation of the catchment into the hydrologically similar unit (HSU) classes that results in a cascade of 10-15 downslope HSUs. This version has since been developed further at Lancaster University and used in Beven et al. (2020). The Regional Hydro-Ecological Simulation System (RHESsys) that had originally combined the BIOME_BGC biogeochemical model with the original version of





Topmodel (Band et al., 1993), also later incorporated a similar gridded kinematic routing algorithm as an alternative, (Tague and Band, 2004).

Another gridded kinematic wave model that was inspired by the original Topmodel is Topkapi (Todini, 1995; Ciarapica and Todini, 2002). This was intended to relax the steady state assumption of Topmodel and include the downslope unsaturated fluxes as a function of water content. It was also aimed at having model parameters that were effectively scale independent by integrating the equations over the grid scale into a cascade of nonlinear reservoirs (Liu and Todini, 2005). The theory underlying Topkapi also resulted in a form of soil-topographic index but most applications have been made using a gridded 685 discretisation of the catchment. Later work added deep percolation and groundwater components (Liu et al., 2005).

## 12. Wider applications of Topmodel

There have been a number of different areas where variants of Topmodel have been used as the hydrological basis for other 690 types of predictions.

The original evapotranspiration and root zone component of Topmodel was very simple. This was by design, so as to again introduce only the minimum number of parameters to be calibrated (only one parameter is needed, the effective available water capacity for actual evapotranspiration). This was supported by the study of Calder et al. (1983) who showed that very simple 695 evapotranspiration models could reproduce soil moisture deficits just as well as complex models as sites across the UK, even during the extreme drought year of 1976. Since then, however, there has been some interest in taking more explicit account of different vegetation covers within the Topmodel framework. Beven and Quinn (1994) used a more complex root zone representation, including the possibility of capillary rise, in studies of variability in water balance (see also Tague and Band, 2004).


New forms were also driven by the aim of incorporating some effects of topographic and vegetation variability into the land surface parameterisations of atmospheric circulation models. The earliest attempt to do so was the Topmodel-based Land surface – Atmosphere Transfer Scheme (Toplats) formulation produced in the PhD of Jay Famiglietti at Princeton University (Famiglietti and Wood, 1994). This was later modified to add more energy budget components (Peters-Lidard et al., 1997; 705 Pauwels and Wood, 1999), and is still being used (e.g. Fu et al., 2018). The potential for allowing for subgrid variability in hydrological states in the TOPUP land surface parameterisation based on Topmodel was also explored by Quinn et al. (1995b) and Franks et al. (1997). A version of Topmodel was later included in the land surface parameterisations used by the UK MetOffice (MOSES2 then JULES, Essery et al., 2003; Best et al., 2011; Zulkafli et al., 2013), while MeteoFRANCE use ISBA-Topmodel as a land surface parameterisation (Habets and Saulnier, 2001; Vincendon et al., 2010). There has also been 710 a form of land surface parameterisation based on Dynamic Topmodel called HYDROBLOCKS developed by Chaney et al.



(2016) designed to allow the representation of high-resolution local variability. We note, however, that simple use of the Topmodel concepts are very unlikely to be valid in many parts of the globe where these land surface parameterisations are likely to be used. It is to be hoped that they are used with care.

Another interesting extension of the use of Topmodel and Dynamic Topmodel has been in the prediction of solute concentrations in small catchments. One of the features of being able to map the predictions back into space is that the pattern of storage deficits or water table levels along the stream network can be determined on a time step by time step basis. Robson et al. (1992) made use of this by assuming, on the basis of field observations that different soil horizons could be associated with different chemical signatures, with the resulting stream concentration being made up of water displaced from those

horizons. Both stream chemistry and the hydrograph separation between near surface acidic and deeper more buffered waters were simulated reasonably well in a small stream in upland Wales. Page et al. (2007) used Dynamic Topmodel to simulate chloride concentrations for two streams at Plynlimon, adding some exchanges with "immobile" storage to account for the differences in flow and tracer responses. Chloride was chosen as a relatively conservative tracer, but it was found that for the period under study the observations had a marked inbalance between inputs and outputs, possible due to dry deposition and

occult deposition in this maritime site. It was therefore necessary to reconstruct the input signal. Additional mixing assumptions and parameters were required for the model stores. It was shown that the model could reproduce the long term seasonal behaviour quite well, but it did not do so well on the short term storm dynamics.

The topographic index has also been used as the basis for mapping of relative risk for solutes, sediments and faecal bacteria

within the SCIMAP system (e.g. Lane et al., 2009; Milledge et al., 2012; Porter et al., 2016).

## 13. What would we do now?

A lot has happened since Topmodel was originally formulated in the 1970s especially in terms of the computer power available

to modellers and information about catchments through mapping and satellite images. Some things, however, remain only rather poorly known; perhaps most importantly the subsurface structures and flow characteristics in catchments, including the potential for preferential flows (Beven and Germann, 1982; 2013) and changing connectivities on hillslopes (Hopp and McDonnell, 2009; Jensco and McGlynn, 2011; Tetzlaff et al., 2014; Bergstrom et al., 2016). Some important new understanding of catchment processes has been gained over that period, particularly the use of environmental isotopes to study

the residence times and contributions of pre-event water to hydrographs. While it is not strictly necessary to take this into account in modelling catchment discharges (see the discussion of velocities and celerities in McDonnell and Beven, 2014, and Beven, 2020, for example), there have been increasing demands to link surface and subsurface predictions to solute transport, sediment mobilisation and transport, and biogeochemistry and many studies have used Topmodel as a basis for building more complex model structures and land surface parameterisations (e.g. HYDROBLOCKS, RHESSys, JULES etc as



noted above). It certainly seems that the simplicity of the topographic index approach is still attractive. Perhaps too attractive, in that it is clear that many applications of Topmodel have been to catchments where the assumptions are clearly not even approximately valid. Regardless of the validity of the assumptions it can still be calibrated to provide a nonlinear runoff generation function, but it might be hoped that applications would be made with a bit more hydrological thought.

It is, however, interesting to speculate about what we would do if we were starting over to develop a simple hydrological model that showed some physical basis to its process representations, including the effect of hillslope form on surface and subsurface runoff generation; that was fast to run so that the uncertainty associated with the predictions could be assessed and/or such models can be run for "everywhere"; and where the spatial predictions could be mapped back into space to give the potential for some evaluation and inference about processes in the catchment. This is already a demanding set of

requirements, satisfied by Topmodel through the use of the topographic index as an index of hydrological similarity but few other models. The key assumption is then that the saturated zone takes up a configuration as if there was a uniform recharge flux everywhere on the hillslopes equivalent to the saturated zone discharge.

Increased computer power does mean that some complexity can be added, while still retaining the possibility of running the

model many times. This is already reflected in the explicit downslope routing incorporated into Dynamic Topmodel, the Distributed Topmodel, and DECIPHeR which has also allowed the relaxation of some of the other assumptions of the Original Topmodel and a more explicit account of spatial heterogeneities that are perceived as being hydrological important. There is still some computational advantage of retaining a similarity idea (as in Dynamic Topmodel) as opposed to a fully distributed model which will normally require a coarser spatial resolution for similar run times.


The process descriptions in Dynamic Topmodel have not, however, changed so very much. There is more flexibility in defining the subsurface transmissivity functions, and storage deficits at which downslope flows cease to allow for dynamic conductivity effects, but the constant local hydraulic gradient, root zone, and recharge calculations have not changed significantly. In part this was always driven by a wish to reduce the number of parameters; including not separating

interception and root zone stores and allowing recharge only when the root zone reaches some "field capacity" threshold. Preferential flows are implicit in this excess for vertical flows and in the transmissivity function for downslope flows. Different options would be relatively simple to implement if driven by a different perceptual model of catchment response or different application requirements as in the different implementations of the original Topmodel, but again at the cost of requiring more parameter values to be defined.


It is possible to think about a different way of process representations in models of this type by drawing an analogy between these HSUs and the Representative Elementary Watershed (REW) concept (e.g. Reggiani and Rientjes, 2005). In the REW framework, the mass balance, energy balance and momentum balance conservation equations should hold at any discretisation



scale. What is then necessary to apply those equations is to relate the boundary fluxes of each element to the internal states of
that element. This approach has the advantage of avoiding any reference to continuum differential equations in systems
where gradients of most variables at larger scale do not really have much meaning. It also implies that, given that there is a
length scale for any discretisation, the process representation that links states and fluxes should be scale dependent (Beven,
2006). Here is an opportunity for real progress to be made in terms of model parameterisations, but the problem has not yet
been solved. How, for example, does the hysteresis in the boundary fluxes relative to states, due to differences between
velocities and celerities, change with scale and wetness (e.g. Davies and Beven, 2015). And, if we are interested in predicting
transport as well as flows, how do the residence times and transit time distributions in an element vary with that hysteresis. It
would be a real advance to make progress with this type of representation in a simple parametrically parsimonious way.
However, any evaluation of the results will continue to depend on also making advances in observational methods (Beven,
2019; Beven et al., 2020).

Another aspect of hydrological modelling that has become more possible with modern computing and databases is running
models to represent "everywhere" (Beven, 2007; Beven and Alcock, 2012; Blair et al., 2019). This has the potential to change
the way that modelling is done from trying to find generalised model structures to apply widely, to a learning process about
places. Dynamic Topmodel, in its DECIPHeR form, has already been applied to the whole of Great Britain (Coxon et al.,
2019) making it very obvious where some of the model assumptions were not valid and local modifications would be necessary
(notably in predicting catchments with large groundwater storages). Other catchments where the assumptions might be
expected to hold better also showed rather variable modelling efficiencies but, again, this might be a matter of uncertainties in
the hydrological data as much as a problem of model structure. What such results do encourage, is the investigation of why
the predictions are not so good locally, and not only at the catchment level but also at very local levels where spatial predictions
can be compared with useful hydrological information. This can include specific monitoring at sites of interest, organised
field campaigns, and citizen science type of local information as part of the learning process. Learning from where the model
predictions can be shown to be wrong is certainly going to be one way to advance modelling practice in future. The flexibility
of the different ways of classifying the landscape and the possibility of modifying model structures and parameters in different
classes in the different implementations of Dynamic Topmodel should prove useful in that respect.

**Acknowledgements**

KB would like to recognise the characteristic generosity of Mike Kirkby in allowing the original author order to be alphabetical,
despite the fact that Topmodel was based on his original idea for the topographic index. Not only that but he even offered his
new post-doc a room in his house while he house-hunted in Leeds, a stay that extended to 3 months! Many people have





contributed to the development of the Topmodel concepts and associated fieldwork over the years. We would like to acknowledge particularly Bruno Ambroise, Daniela Balin, Kathy Bashford, Sarka Blazkova, Patricia Bruneau, Wouter
Buytaert, Paul Campling, Flavie Cernesson, Pierre Chevalier, Hyung Tae Choi, Roger Clapp, Jack Cosby, Gemma Coxon, Patrick Durant, James Fisher, Stewart Franks, Frances Gallart, Chantal Gascuel-Odoux, Anna Gobin, Hannah Green, George Hornberger, Ion Iorgulescu, Dick Iredale, Christophe Joerin, Alena Kulusova, Jerome Latron, Pilar, Llorens, Yanli Liu, Jeff McDonnell, Hilary McMillan, Philippe Merot, Norm Miller, Stefan Myrabø, Colin Neal, Charles Obled, Bo Ostendorf, Trevor Page, Jake Peters, Pep Piñol, Olivier Planchon, N. Pradhan, Paul Quinn, Alice Robson, Renata Romanowicz, David
Sappington, Georges-Marie Saulnier, Jan Seibert, Murugesu Sivapalan, Nick Schofield, Paul Smith, Andy Tagg, Jacques Wendling, Eric Wood, and Philip Younger. We would also like to dedicate the paper to the memory of Peter Metcalfe, who was applying a version of Dynamic Topmodel in R to natural flood management applications with great enthusiasm and effectiveness at the time of his death in a climbing accident in 2018. The Q-NFM project and the contribution of KB to this paper was funded by NERC Grant NE/R004722/1 led by Nick Chappell

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
