# Peer review of "A history of TOPMODEL"

_Hydrology and Earth System Sciences, 2020_

## Referee Comment (RC1) · Francesc Gallart (Referee) · 31 Aug 2020

First of all, I want to congratulate the authors and appreciate their keenness in writing this nice article. It is sensibly short and easy to read and, although I already knew part of this history, I learned or consolidated many aspects of it published in media not always easy to get. The manuscript includes not only most of the TOPMODEL developments but also several criticisms it received during this already long history, in a frank and friendly style.

I have some suggestions to include or enlarge comments on issues that arise from my experiences. I also include a list of typos to be corrected.

- Scale dependence of the topographic index. This issue is mentioned in line 375, but the writing used is not sufficiently explicit, so readers that do not know this effect might miss the message.

[Figure]

- Topographic index used with too large areas or mesh sizes. The topographic index is designed for hillslope hydrology, so when it is applied to an area large enough to include channels, it may take very high values that fall out of its intended physical sense (Quinn et al., 1995a). Likewise, when the index is calculated with a mesh size that is large enough to include channels, the physical sense is also lost. These issues seem to be indirectly mentioned after line 745, but I deem that they deserve a more explicit comment.

- Negative values of deficit. During wet periods, local saturation deficits may take negative values in saturated areas and even the mean catchment deficit may become negative (Saulnier and Datin, 2004). Although this behaviour is not an issue for the application of the model, it compromises its physical soundness and therefore should deserve some comment in the paper.

- Value of TOPMODEL for teaching. While I understand that this is not the purpose of the paper, I wonder if some comment on TOPMODEL value for teaching could be included. I agree that its relative simplicity means that it can be riskily used as passe-partout conceptual scheme for areas that do not meet the key assumptions, as it happened with the models based on the precipitation excess process. But if this risk is bounded, according to my experience, one of the strengths of TOPMODEL for teaching is its usefulness to exemplify the model parameter compensation and equifinality issues, as well as the consequent principle that a good flow efficiency of a model does not mean that it works for the right reasons. The paragraph starting at line 525 is a superb example of these questions, how field observations can be used for their amendment and the lessons that not just KB but every one of us can learn.

Minor comments:

- Line 149: the abbreviation BK79 is used before it is defined in line 182.

- Line 298: "Rigelbach" should be Ringelbach.

[Figure]

- Line 649: "Staring" should be starting.

- Line 777: "Disretisation" should be discretisation.

- Line 816: "Frances" should be Francesc

- Line 817: "Kulusova, Jerome Latron, Pilar, Llorens" should be Kulasova, Jérôme Latron, Pilar Llorens.

- Line 1089: "Leibendgut" should be Leibundgut.

- Line1324: "Bloschl" should be Blöschl

---

## Short Comment (SC1) · 31 Aug 2020

Dear Auhors, thanks for this review !

Also maybe of interest for the historical perspective:

- based on the analytical study of the "bias" mentionned in Saulnier and Datin 2004 we introduced in the TOPMODEL formulation, before coupling with a SVAT model over large basins, a critical deficit above which lateral flow is assumed to be negligible (cf. line 348, see Appendix 2 in Boulet, G., Kerr, Y., Chehbouni, A., and Kalma, J. D.: Deriving catchment-scale water and energy balance parameters using data assimilation based on extended Kalman filtering, Hydrological Sciences Journal-Journal Des Sciences Hydrologiques, 47, 449-467, 2002);

- TOPMODEL is/was a good candidate for disaggregation stragegies of low resolution

soil moisture data from passive microwave remote sensing (e.g. Pellenq, J., Kalma, J., Boulet, G., Saulnier, G. M., Wooldridge, S., Kerr, Y., and Chehbouni, A.: A dis-aggregation scheme for soil moisture based on topography and soil depth, Journal of Hydrology, 276, 112-127, 10.1016/s0022-1694(03)00066-0, 2003);

---

## Referee Comment (RC2) · Dave Milledge (Referee) · 10 Sep 2020

I enjoyed reading this paper and found it both interesting and informative. It is different from the papers that I am used to reading because it reads more as a careful and balanced reflection on a model rather than a report of new findings. But I think it is valuable and will be a useful resource to those who use or are considering using Topmodel in the future as well as those who must make sense of its predictions. I have four major comments, none of which are critical to publication but all of which I feel would improve the paper. There are then minor comments and suggestions most of which are either typos or suggested rewording in the attached pdf.

Major comments

Assumption A1

[Figure]

The assumption that "that the storage for any given value of Sbar is configured as if it was at a steady state with a steady homogeneous recharge rate (L88)", and its implications comes up in three different places within the article. It is an important point because it relates to a central assumption and one of the primary perceived weaknesses of Topmodel. I found this discussion particularly helpful in my thinking on Topmodel but I also found it confusing in places.

On my first read through I felt the first discussion of A1 on L88 didn't give enough detail. In particular I was confused by the language around "configurations" and "configured as if". I didn't understand how Sbar could be configured as if it was at steady state (L88) nor how configurations are dependent on storage (L89) nor how the two ideas related to one another. Did this mean that Sbar is varying only slowly in time? How slowly does it need to be? What controls the sensitivity of Sbar to rainfall and what is the sensitivity of the saturated zones to Sbar?

The later treatment of the assumption (L320) is more detailed and I understood this section better. It might be enough just to point to the later section at L89 for more detail. In this L320 paragraph I still struggled to understand what you meant by configuration. I understood it to mean that: 'the two-dimensional phreatic surface over the flow strip is that which would result from steady recharge over that flow strip'. However I wasn't confident in my understanding so clarifying this would be helpful. The main outstanding question for me at the end of the paragraph was: how close to 'as if' is near enough? You mention this with reference to Kirkby (1997) but a more complete restatement of his examination and findings would be useful here.

Assumption A1 is revisited on L755, and I found this the clearest expression of the steady state assumption within the paper. It may be that the other sections had laid the groundwork but I think you should consider re-stating this expression earlier in the paper.

Assumption A2
Topmodel uses tan(beta) to calculate lateral subsurface flux (L49 and equation 1). Others, usually modelling steep landscapes, have used sin(beta) to make the same calculation (e.g. Montgomery and Dietrich, 1994; 2002; Borga et al., 2002; Chirico et al., 2003). In some cases they explicitly claim that there is a choice between "the original ln(A/tan(beta)) or the more physically correct ln(A/sin(beta))" (Montgomery and Dietrich, 2002, p2). It might be helpful to respond to this claim, perhaps explaining why the difference, whether you consider one more physically correct than the other and if so what the implications are for situations in which they can or should be applied.

Assumption A3

It would be useful to have a longer discussion of whether the exponential transmissivity function is an assumption introduced by the authors (as is suggested L91-2) or one that is required within the derivation (as Kirkby (1997) seems to suggest). There are clearly advantages to being able to use alternative transmissivity functions, so it would be useful to know more about any possible disadvantages. It would be particularly useful to comment on how this might impact the validity of other model assumptions (e.g. L413) and the sensitivity to these assumptions (e.g. L360-2)

You do touch on this at L413 "might also preclude..." however, you say might rather than would and I am not clear what you mean by "implicit redistribution of subsurface storage". Do you mean that A1 would not be consistent with non-exponential transmissivity functions? Kirkby (1997) seems to argue that the choice of an exponential transmissivity function is required to satisfy the integration (though I could have misunderstood Kirkby here). Do the authors of this paper find that argument convincing? If so what does it mean for the alternative profiles (e.g. Ambrose et al., 1996)? If not then where do you differ from Kirkby (1997)?

Connectivity and run on

The argument that small channels may connect apparently disconnected saturated areas (L350) is not clear to me. In particular mention of small channels at the start of
the sentence seems to contradict the end of the sentence. If I understand what you mean here, I think it might be clearer to talk about geomorphic / landscape evolution controls on where channels begin (e.g. Montgomery and Dietrich, 1988). The places where this will break down are those where some other landscape property gets in the way e.g. lithology and rock strength in parts of the Yorkshire Dales. If instead this is a suggestion that the majority of run-on passes from patch to patch and reaches the river as overland flow, then I think more support for the argument is needed. I haven't seen anyone demonstrate this.

References

Borga, M., Dalla Fontana, G. and Cazorzi, F., 2002. Analysis of topographic and climatic control on rainfall-triggered shallow landsliding using a quasi-dynamic wetness index. Journal of Hydrology, 268(1-4), pp.56-71. Chirico, G.B., Grayson, R.B. and Western, A.W., 2003. On the computation of the quasi‐dynamic wetness index with multiple‐flow‐direction algorithms. Water resources research, 39(5). Montgomery, D.R. and Dietrich, W.E., 1994. A physically based model for the topographic control on shallow landsliding. Water resources research, 30(4), pp.1153-1171. Montgomery, D.R. and Dietrich, W.E., 2002. Runoff generation in a steep, soil‐mantled landscape. Water Resources Research, 38(9), pp.7-1.

Please also note the supplement to this comment:
https://hess.copernicus.org/preprints/hess-2020-409/hess-2020-409-RC2-supplement.pdf
* * *
[Figure]

**Supplement:**

**Review of Beven et al. 2020 for HESS by David Milledge**

I enjoyed reading this paper and found it both interesting and informative. It is different from the papers that I am used to reading because it reads more as a careful and balanced reflection on a model rather than a report of new findings. But I think it is valuable and will be a useful resource to those who use or are considering using Topmodel in the future as well as those who must make sense of its predictions. I have four major comments, none of which are critical to publication but all of which I feel would improve the paper. There are then many minor comments and suggestions most of which are either typos or suggested rewording.

**Major comments**
**Assumption A1**
The assumption that "that the storage for any given value of Sbar is configured _as if_ it was at a steady state with a steady homogeneous recharge rate (L88)", and its implications comes up in three different places within the article. It is an important point because it relates to a central assumption and one of the primary perceived weaknesses of Topmodel. I found this discussion particularly helpful in my thinking on Topmodel but I also found it confusing in places.

On my first read through I felt the first discussion of A1 on L88 didn't give enough detail. In particular I was confused by the language around "configurations" and "configured as if". I didn't understand how Sbar could be configured as if it was at steady state (L88) nor how configurations are dependent on storage (L89) nor how the two ideas related to one another. Did this mean that Sbar is varying only slowly in time? How slowly does it need to be? What controls the sensitivity of Sbar to rainfall and what is the sensitivity of the saturated zones to Sbar?

The later treatment of the assumption (L320) is more detailed and I understood this section better. It might be enough just to point to the later section at L89 for more detail. In this L320 paragraph I still struggled to understand what you meant by configuration. I understood it to mean that: 'the two-dimensional phreatic surface over the flow strip is that which would result from steady recharge over that flow strip'. However I wasn't confident in my understanding so clarifying this would be helpful. The main outstanding question for me at the end of the paragraph was: how close to 'as if' is near enough? You mention this with reference to Kirkby (1997) but a more complete restatement of his examination and findings would be useful here.

Assumption A1 is revisited on L755, and I found this the clearest expression of the steady state assumption within the paper. It may be that the other sections had laid the groundwork but I think you should consider re-stating this expression earlier in the paper.

**Assumption A2**
Topmodel uses tan(beta) to calculate lateral subsurface flux (L49 and equation 1). Others, usually modelling steep landscapes, have used sin(beta) to make the same calculation (e.g. Montgomery and Dietrich, 1994; 2002; Borga et al., 2002; Chirico et al., 2003). In some cases they explicitly claim that there is a choice between "the original ln(A/tan(beta)) or the more physically correct ln(A/sin(beta))" (Montgomery and Dietrich, 2002, p2). It might be helpful to respond to this claim, perhaps explaining why the difference, whether you consider one more physically correct than the other and if so what the implications are for situations in which they can or should be applied.

**Assumption A3**
It would be useful to have a longer discussion of whether the exponential transmissivity function is an assumption introduced by the authors (as is suggested L91-2) or one that is required within the derivation (as Kirkby (1997) seems to suggest). There are clearly advantages to being able to use alternative transmissivity functions, so it would be useful to know more about any possible disadvantages. It would be particularly useful to comment on how this might impact the validity of other model assumptions (e.g. L413) and the sensitivity to these assumptions (e.g. L360-2)

You do touch on this at L413 "might also preclude…" however, you say might rather than would and I am not clear what you mean by "implicit redistribution of subsurface storage". Do you mean that A1 would not be consistent with non-exponential transmissivity functions? Kirkby (1997) seems to argue that the choice of an exponential transmissivity function is required to satisfy the integration (though I could have misunderstood Kirkby here). Do the authors of this paper find that argument convincing? If so what does it mean for the alternative profiles (e.g. Ambrose et al., 1996)? If not then where do you differ from Kirkby (1997)?

**Connectivity and run on**

The argument that small channels may connect apparently disconnected saturated areas (L350) is not clear to me. In particular mention of small channels at the start of the sentence seems to contradict the end of the sentence. If I understand what you mean here, I think it might be clearer to talk about geomorphic / landscape evolution controls on where channels begin (e.g. Montgomery and Dietrich, 1988). The places where this will break down are those where some other landscape property gets in the way e.g. lithology and rock strength in parts of the Yorkshire Dales. If instead this is a suggestion that the majority of run-on passes from patch to patch and reaches the river as overland flow, then I think more support for the argument is needed. I haven't seen anyone demonstrate this.

**Minor comments**

L29: parsimonious which – word missing

L35: suggesting … constant - has this observation now been more widely reproduced? A comment on this would be useful here.

L50: transmissivity - i.e. the depth integrated permeability? It might help to say this explicitly.

L50: represented … units - This is not clear to me: I think that K is a profile averaged permeability. However, for the following equations to work I think that K must be independent of S and therefore independent of depth. I think this is worth saying here.

Eqn 2: S appears on both the RHS and the LHS. I think that is a typo.

L57: just saturated - just seems a strange word here. Do you mean at the onset of saturation (implying that the timing matters) or the position of the water table (i.e. fully saturated or saturated so that the water table is at the surface)?

Eqn 3: If the soil is "just saturated" shouldn't $S=S_0$? Also, $S_0$ hasn't been defined.

Eqn 4: Kbar and A are not defined yet. I think that Kbar is catchment averaged permeability i.e. $1/A*integral(K,dA)$. If that is the case it seems strange to write it this way because Kbar in lambda cancels the Kbar in the denominator of the equation for Sbar.

L50: K is defined as a permeability but I think for the dimensions to work it should be hydraulic conductivity here and elsewhere.

L68: these equations – this would be clearer as 'equations 3 and 4'.

L89: storage changes - comma needed after changes

L90: wet, … moderate slopes - an indication of the range for these might be helpful if it were possible.

L91: where soil permeabilities increase with saturation - do you mean transmissivities due to hydraulic conductivity decreasing with depth or that permeability (hydraulic conductivity) itself is changing?

L100: just saturated – as for L57.

L102: as – I suggest 'saturation now is:' or 'saturation as:'

L113: Qb … along the channel - It is not clear from this whether Qb is a single value assumed to be distributed along the channel but without explicitly accounting for this distribution or whether it is spatially variable.

L114: see Beven - I think it is fine to point to this more detailed derivation, though it is probably worth adding a page reference (p214). The one difficulty I have there is in Equation B6.1.16 where upslope area per unit contour length is used but this cannot be true for anything other than the outlet cell. I think that a in B6.1.16 should instead be defined as the unique upslope area (that does not overlap with upslope area for any other channel length). This is very intuitive when a channel is viewed as a line (i.e. inputs from the banks of a river) but not as an area with an upslope boundary defined by a contour (when inputs from upstream as well as from the banks should be included).

L135: calculated outputs - Is there a reference associated with this? Fine if not but useful if so.

L149: both showed changes over time - I'm not clear, do you mean that the components have changed over time or that the represented time variable processes?

L151: later these stores were integrated – this would benefit from a reference.

L163: later versions of Topmodel – this would benefit from a reference.

L217: The BK79 paper … flashy catchment – this sentence doesn't make sense to me.

L234: Whilst… - this sentence reads as a fragment.

L235: but were always clearly identified – I think this relates to the assumptions but it is not clear, perhaps there is a comma missing.

L256: including – should be included

L285: but t more – typo, remove t

L332-3: as the catchment wets and dries – An additional sentence explaining why would be helpful here

L336-7: This should not be a surprise at Tarrawarra … impermeable subsoil – this would benefit from a reference

L346: connectivity of both surface and subsurface flows - It seems as though there are two issues being discussed here, disconnection and response timescale. 1) Even if the signal can propagate rapidly in the presence of a water

table configured as assumed by Topmodel it may not propagate if that assumed configuration breaks down resulting partial or total disconnection of some upslope area. 2) Even if the water table is configured as assumed in topmodel the signal propagation may not be sufficient to generate steady state like response to rainfall. I thought Barling's modification was largely focussed on the second problem.

L348: This paragraph, and particularly this sentence contains two ideas that might be more easily understood if separated: subsurface and overland flow connectivity.

L348: always connectivity – do you mean of subsurface flow?

L353: can be represented in… - It has also been represented in the network index, a modification to the topographic index to identify areas of disconnected saturation within Topmodel, the runoff from which is then assumed to entirely reinfiltrate as run-on (Lane et al., 2004, 2009, Lane and Milledge, 2013).

L355: wetter vegetation patterns – perhaps a reference here?

L361: particularly – this was not clear to me. Do you mean that Kirkby shows it for the particular case of exponential transmissivity or that it is particularly true for the exponential transmissivity profile case? I think it is the latter. If so, what are the implications for other assumed transmissivity profiles?

L365: parallel to the surface - How much empirical support is there for assumption A2? You provide a statement on this for A1 and a similar statement here would be useful.

L366: important - It might be helpful to say important to what. This may be tricky given the wide range of applications of Topmodel and its derivatives and I don't think an exhaustive list of 'important to...' is necessary here. However, an indication that both the scale over which the assumption is not valid and the question being addressed contribute to whether or not violation of A2 will be important. For example, I could imagine bedrock exfiltration being very important to local patterns of saturation deficit on the scale of metres to tens of metres (and therefore to pore pressure at scales associated with landslide triggering) but much less important over scales of hundreds to thousands of metres and therefore for runoff generation.

L370: (it might… values) – Has anyone done this? If not it might be worth flagging that more strongly. One of the valuable things about this paper is the opinions of the original authors on what has been done and what is yet to be done from their perspectives.

L374: values tend to be high - Is that a commensurability issue associated with scale dependence in the measurement? If so it might be useful to comment on that here.

L388: A further criticism… unsaturated zone – a reference is needed here. Also, this doesn't seem to get the detailed response that other criticisms received. Is the following sentence a general conclusion to the section or specific to this criticism?

L425: small – adding the catchment area would be useful here.

L429-30: This was … number of storms – mixed tenses in this sentence

L431: would predict – should this be: would predict (i.e. it is possible) or predicted (i.e. this was the prediction)?

L463: predicted modelled – I think only one or the other of predicted or modelled is needed here.

L463: storage deficits to water tables - You don't give this much treatment. Can you point to a reference and give an indication of the magnitude of the issue, as written it appears a straightforward exercise.

L466: represent different quantities - An example here might be helpful. E.g. modelled water table depth at 2 m resolution compared with observations from a single well within that 2 m cell.

L467: this is a particular problem - Is it commensurability between predictions and observations that is a particular problem in this case or commensurability between the input parameter (spatially uniform T0) and the local conditions (heterogeneous hydraulic conductivity in 3-D)?

L470: would not expect the predictions to match… - Has anyone provided an indication of the likely magnitude of the commensurability errors? If so how do these compare to the misfit observed in these studies?

L505: do not explicitly consider convergence… - I disagree: the indices of Barling, Woods, explicitly consider convergence etc through upslope area as the topographic index does, that of Berne uses a hillslope width function and that of Hjerdt was conceived as a replacement for local slope which has subsequently been used within a modified a/tanB topographic index.

L521: a rumour - this moves away from the authors' own experiences and is difficult to support. Consider rephrasing or removing the last two or three sentences since I think you can make your point without this statement.

L528: that contributing – should be that the contributing

L601-5: One of the… A second result… - A reference that you associate with each of these results would be helpful at the end of each sentence.

L605: longer time steps – could you quantify longer timesteps?

L626: risk - Could you expand this to replace 'risk'? You haven't previously defined risk and it may mean different things to different people.

L646: A further step - Should this be 'an alternative step'? Dynamic Topmodel predates the distributed version and does not adopt its surface flow treatment.

L696-7: some interest - references to this interest would be useful here, unless the following sentence is an example.

L729: topographic index - this actually uses the network index, a modified version of the topographic index that accounts for downslope connectivity of saturation excess overland flow (see: Lane et al., 2004; 2009).

L761: some of the other assumptions – it would be useful to be explicit about which assumptions here.

L762: hydrological important – should be: hydrologically important.

L767: cease to allow – I think there is a comma missing here.

L780: This approach … much meaning – it is not clear to me what this comment means.

**References**

Borga, M., Dalla Fontana, G. and Cazorzi, F., 2002. Analysis of topographic and climatic control on rainfall-triggered shallow landsliding using a quasi-dynamic wetness index. *Journal of Hydrology*, *268*(1-4), pp.56-71.

Chirico, G.B., Grayson, R.B. and Western, A.W., 2003. On the computation of the quasi-dynamic wetness index with multiple-flow-direction algorithms. *Water resources research*, *39*(5).

Lane, S.N., Brookes, C.J., Kirkby, M.J. and Holden, J., 2004. A network-index-based version of TOPMODEL for use with high-resolution digital topographic data. *Hydrological processes*, *18*(1), pp.191-201.

Lane, S.N., Reaney, S.M. and Heathwaite, A.L., 2009. Representation of landscape hydrological connectivity using a topographically driven surface flow index. *Water Resources Research*, *45*(8).

Lane, S.N. and Milledge, D.G., 2013. Impacts of upland open drains upon runoff generation: a numerical assessment of catchment-scale impacts. *Hydrological Processes*, *27*(12), pp.1701-1726.

Montgomery, D.R. and Dietrich, W.E., 1988. Where do channels begin?. *Nature*, *336*(6196), pp.232-234.

Montgomery, D.R. and Dietrich, W.E., 1994. A physically based model for the topographic control on shallow landsliding. *Water resources research*, *30*(4), pp.1153-1171.

Montgomery, D.R. and Dietrich, W.E., 2002. Runoff generation in a steep, soil-mantled landscape. *Water Resources Research*, *38*(9), pp.7-1.

---

## Referee Comment (RC3) · Wouter Buytaert (Referee) · 20 Oct 2020

Wouter Buytaert (Referee)

w.buytaert@imperial.ac.uk

This is a very nice overview of the history of Topmodel and I think that it will serve as an extremely useful reference for anyone who may feel a bit lost in the numerous variations of model structures that have emerged over the year (which at least was the case with this reviewer).

I don't have much to add to the other reviews apart from a few specific comments (see below); however I would like to second Dave Milledge's comments on the lack of discussion on landscape connectivity in determining the contribution of surface runoff to river flow, as explored in publications such as Lane et al., (2004) and Lane et al., (2009). At least, looking back on my own applications of Topmodel, I identify our inability to account properly for varying surface contributing area as a result of dynamic surface hydrological connectivity as a major bottleneck for our model performance (in addition

to errors in the input data). Even though this process could clearly be observed in the field, there was no way to incorporate this in the model (apart from some simple and unsatisfactory conceptual approaches) because of the lack of topographic information of sufficient quality. However, recent advances in drone-based remote sensing, including the generation of cm-resolution digital elevation models, may open interesting new opportunities in this regard and I would be very interested in hearing the authors' views on this.

(On the other end of topographic-data-availability-spectrum, I seem to recall some successful implementations of Topmodel that bypassed the topographic index derivation altogether in favour of the use of a gamma distribution with calibrated parameters.)

Specific comments

l55: there is an error in the formula (S on the right hand side should not be there)

l60: it may be useful to define $S_0$ here explicitly - also further in the document, not all symbols are always clearly defined

l237: ofa -> of

l249: attractive. -> attractive

l285: "but t more"

l452: "runoff." -> "runoff"

l730: Lane et al. (2009) is referenced here in relation to SCIMAP although the publication is not really about SCIMAP.

References:

Lane, S. N., Brookes, C. J., Kirkby, M. J., & Holden, J. (2004). A network-index-based version of TOPMODEL for use with high-resolution digital topographic data. Hydrological Processes, 18, 191–201.

Lane, S. N., Reaney, S. M., & Heathwaite, A. L. (2009). Representation of landscape hydrological connectivity using a topographically driven surface flow index. Water Resources Research, 45, W08423.

---

## Author Comment (AC1) · 21 Oct 2020

Thanks for the useful comments Wouter. They will be incorporated into the revision of the paper, especially in strengthening the discussion of connectivity as one reason for the introduction of Dynamic Topmodel.

K

---

## Author Comment (AC2) · 29 Oct 2020

Thanks for the comment, Gilles.

I was not aware of these papers.

k

---

## Author Comment (AC3) · 29 Oct 2020

Thanks for the very useful comments Francesc.

We shall expand on the points you mention, including the value in teaching (which was, I think helped by making the teaching version freely available), and make the corrections you have noted.

k
* * *

---

## Author Comment (AC4) · 29 Oct 2020

Thanks for the careful reading and useful comments, Dave. We will take your points into account when revising the manuscript. The comments on Assumption A1 are particularly valuable and will be used to try and make this clearer earlier in the paper.

I had a long discussion about Assumption A2 Wirth Dave Montgomery in the field at his Coos Bay site many years ago. SinB is physically more correct if the slope distance is measured along the slope, and transmissivity in the direction of flow. TanB corrects for water balance in using plan distance (for a given input rate) but then requires an effective transmissivity value. Since transmissivity is in any case a rather uncertain parameter and the differences become significant only at larger slope angles, the effective nature of the transmissivity parameter is implicit in the calibration.

We can agree with the comment about run-on. The argument is a consequence of

field observations in upland UK where areas that produce surface runoff frequently will often have a connection to small rivulets or streams, albeit often subtly revealed in the topography and vegetation, and at scales smaller than the grid size. This is rather different to the Montgomery and Dietrich channel head arguments (albeit that the topographic index has proven rather useful there). I certainly would not argue for the model representing the idea of overland flow as a sheet flow in producing run-on - hence the comments in the paper. We will try to express this more clearly.

Thanks also for the additional corrections and suggestions in the mss.

k

---

## Author Response (AR1)

**Response to Referees (and additional comment in HESSD)**

Note that Line Numbers refer to the version with Tracked Changes rather than the clean version.

**Francesc Gallart (Referee)**

francesc.gallart@idaea.csic.es

First of all, I want to congratulate the authors and appreciate their keenness in writing this nice article. It is sensibly short and easy to read and, although I already knew part of this history, I learned or consolidated many aspects of it published in media not always easy to get. The manuscript includes not only most of the TOPMODEL developments but also several criticisms it received during this already long history, in a frank and friendly style.

I have some suggestions to include or enlarge comments on issues that arise from my experiences. I also include a list of typos to be corrected.

- Scale dependence of the topographic index. This issue is mentioned in line 375, but the writing used is not sufficiently explicit, so readers that do not know this effect might miss the message.

- Topographic index used with too large areas or mesh sizes. The topographic index is designed for hillslope hydrology, so when it is applied to an area large enough to include channels, it may take very high values that fall out of its intended physical sense (Quinn et al., 1995a). Likewise, when the index is calculated with a mesh size that is large enough to include channels, the physical sense is also lost. These issues seem to be indirectly mentioned after line 745, but I deem that they deserve a more explicit comment.

The discussion of this has now been expanded in a paragraph starting at L398.

- Negative values of deficit. During wet periods, local saturation deficits may take negative values in saturated areas and even the mean catchment deficit may become negative (Saulnier and Datin, 2004). Although this behaviour is not an issue for the application of the model, it compromises its physical soundness and therefore should deserve some comment in the paper.

Text has been added starting at L142ff.

- Value of TOPMODEL for teaching. While I understand that this is not the purpose of the paper, I wonder if some comment on TOPMODEL value for teaching could be included. I agree that its relative simplicity means that it can be riskily used as passe-partout conceptual scheme for areas that do not meet the key assumptions, as it happened with the models based on the precipitation excess process. But if this risk is bounded, according to my experience, one of the strengths of TOPMODEL for teaching is its usefulness to exemplify the model parameter compensation and equifinality issues, as well as the consequent principle that a good flow efficiency of a model does not mean that it works for the right reasons. The paragraph starting at line 525 is a superb example of these questions, how field observations can be used for their amendment and the lessons that not just KB but every one of us can learn.

A paragraph has been added at L329

Minor comments:
- Line 149: the abbreviation BK79 is used before it is defined in line 182.   Done

- Line 298: "Rigelbach" should be Ringelbach.  Done

- Line 649: "Staring" should be starting.  Done

- Line 777: "Disretisation" should be discretisation.  Done

- Line 816: "Frances" should be Francesc  Done

- Line 817: "Kulusova, Jerome Latron, Pilar, Llorens" should be Kulasova, Jérôme Latron, Pilar Llorens.  Done

- Line 1089: "Leibendgut" should be Leibundgut.  Done

- Line1324: "Bloschl" should be Blöschl  Done

**Dave Milledge (Referee)**

david.milledge@newcastle.ac.uk

I enjoyed reading this paper and found it both interesting and informative. It is different from the papers that I am used to reading because it reads more as a careful and balanced reflection on a model rather than a report of new findings. But I think it is valuable and will be a useful resource to those who use or are considering using Topmodel in the future as well as those who must make sense of its predictions. I have four major comments, none of which are critical to publication but all of which I feel would improve the paper. There are then many minor comments and suggestions most of which are either typos or suggested rewording.

**Major comments**
**Assumption A1**
The assumption that "that the storage for any given value of Sbar is configured *as if* it was at a steady state with a steady homogeneous recharge rate (L88)", and its implications comes up in three different places within the article. It is an important point because it relates to a central assumption and one of the primary perceived weaknesses of Topmodel. I found this discussion particularly helpful in my thinking on Topmodel but I also found it confusing in places.
On my first read through I felt the first discussion of A1 on L88 didn't give enough detail. In particular I was confused by the language around "configurations" and "configured as if". I didn't understand how Sbar could be configured as if it was at steady state (L88) nor how configurations are dependent on storage (L89) nor how the two ideas related to one another. Did this mean that Sbar is varying only slowly in time? How slowly does it need to be? What controls the sensitivity of Sbar to rainfall and what is the sensitivity of the saturated zones to Sbar?
The later treatment of the assumption (L320) is more detailed and I understood this section better. It might be enough just to point to the later section at L89 for more detail. In this L320 paragraph

I still struggled to understand what you meant by configuration. I understood it to mean that: 'the two-dimensional phreatic surface over the flow strip is that which would result from steady recharge over that flow strip'. However I wasn't confident in my understanding so clarifying this would be helpful. The main outstanding question for me at the end of the paragraph was: how close to 'as if' is

near enough? You mention this with reference to Kirkby (1997) but a more complete restatement of his examination and findings would be useful here.

Wording changed to avoid configuration. More reference to the Kirkby paper has been added later in relation to alternative transmissivity functions (L115-130, L487).

Assumption A1 is revisited on L755, and I found this the clearest expression of the steady state assumption within the paper. It may be that the other sections had laid the groundwork but I think you should consider re-stating this expression earlier in the paper.

??? Not sure about this comment as this section only discusses relaxing A1 in Dynamic Topmodel

**Assumption A2**
Topmodel uses tan(beta) to calculate lateral subsurface flux (L49 and equation 1). Others, usually modelling steep landscapes, have used sin(beta) to make the same calculation (e.g. Montgomery and Dietrich, 1994; 2002; Borga et al., 2002; Chirico et al., 2003). In some cases they explicitly claim that there is a choice between "the original ln(A/tan(beta)) or the more physically correct ln(A/sin(beta))" (Montgomery and Dietrich, 2002, p2). It might be helpful to respond to this claim, perhaps explaining why the difference, whether you consider one more physically correct than the other and if so what the implications are for situations in which they can or should be applied.

A paragraph has been added to discuss this (L68ff)

**Assumption A3**
It would be useful to have a longer discussion of whether the exponential transmissivity function is an assumption introduced by the authors (as is suggested L91-2) or one that is required within the derivation (as Kirkby (1997) seems to suggest).

See expanded text (L115-130, 487)

There are clearly advantages to being able to use alternative transmissivity functions, so it would be useful to know more about any possible disadvantages. It would be particularly useful to comment on how this might impact the validity of other model assumptions (e.g. L413) and the sensitivity to these assumptions (e.g. L360-2)
You do touch on this at L413 "might also preclude…" however, you say might rather than would and I am not clear what you mean by "implicit redistribution of subsurface storage". Do you mean that A1 would not be consistent with non-exponential transmissivity functions? Kirkby (1997) seems to argue that the choice of an exponential transmissivity function is required to satisfy the integration (though I could have misunderstood Kirkby here). Do the authors of this paper find that argument convincing? If so what does it mean for the alternative profiles (e.g. Ambrose et al., 1996)? If not then where do you differ from Kirkby (1997)?

**Connectivity and Run-on**
The argument that small channels may connect apparently disconnected saturated areas (L350) is not clear to me. In particular mention of small channels at the start of the sentence seems to contradict the end of the sentence. If I understand what you mean here, I think it might be clearer to talk about geomorphic / landscape evolution controls on where channels begin (e.g. Montgomery and Dietrich, 1988). The places where this will break down are those where some other landscape property gets in the way e.g. lithology and rock strength in parts of the Yorkshire Dales. If instead this is a suggestion that the majority of run-on passes from patch to patch and reaches the river as

overland flow, then I think more support for the argument is needed. I haven't seen anyone demonstrate this.

The discussion of connectivity and run-on has been extended (see L449ff)

**Minor comments**
L29: parsimonious which – word missing   Done
L35: suggesting ... constant - has this observation now been more widely reproduced? A comment on this would be useful here.  Comment added
L50: transmissivity - i.e. the depth integrated permeability? It might help to say this explicitly.  Done
L50: represented … units - This is not clear to me: I think that K is a profile averaged permeability. However, for the following equations to work I think that K must be independent of S and therefore independent of depth. I think this is worth saying here.   Made more explicit
Eqn 2: S appears on both the RHS and the LHS. I think that is a typo. Corrected
L57: just saturated - just seems a strange word here. Do you mean at the onset of saturation (implying that the timing matters) or the position of the water table (i.e. fully saturated or saturated so that the water table is at the surface)?  Modified to indicate just saturated to the surface
Eqn 3: If the soil is "just saturated" shouldn't S=S_0? Also, S_0 hasn't been defined.  Corrected
Eqn 4: Kbar and A are not defined yet. I think that Kbar is catchment averaged permeability i.e. 1/A*integral(K,dA). If that is the case it seems strange to write it this way because Kbar in lambda cancels the Kbar in the denominator of the equation for Sbar.   This is carried over from the original Kirkby presentation – but has now been modified to explain this form.
L50: K is defined as a permeability but I think for the dimensions to work it should be hydraulic conductivity here and elsewhere.   Not exactly because of the per unit storage rather than per unit depth definition.   This has now been made explicit on L50
L68: these equations – this would be clearer as 'equations 3 and 4'.  Done
L89: storage changes - comma needed after changes Done
L90: wet, … moderate slopes - an indication of the range for these might be helpful if it were possible.  See expanded discussion L72
L91: where soil permeabilities increase with saturation - do you mean transmissivities due to hydraulic conductivity decreasing with depth or that permeability (hydraulic conductivity) itself is changing?  Note definition of permeability earlier – changed to depth of saturation
L100: just saturated – as for L57.   Modified as before
L102: as – I suggest 'saturation now is:' or 'saturation as:' Done
L113: Qb … along the channel - It is not clear from this whether Qb is a single value assumed to be distributed along the channel but without explicitly accounting for this distribution or whether it is spatially variable. ???? It is twice stated that this is the output integrated along the channel. Changed to all channel reaches.
L114: see Beven - I think it is fine to point to this more detailed derivation, though it is probably worth adding a page reference (p214).  Done  The one difficulty I have there is in Equation B6.1.16 where upslope area per unit contour length is used but this cannot be true for anything other than the outlet cell. I think that a in B6.1.16 should instead be defined as the unique upslope area (that does not overlap with upslope area for any other channel length). This is very intuitive when a channel is viewed as a line (i.e. inputs from the banks of a river) but not as an area with an upslope boundary defined by a contour (when inputs from upstream as well as from the banks should be included).   These are in fact equivalent because the a values can be defined on a point by point basis along the channel reaches.   Thus the integral of $l_j a_j$ over all points does give the total area.   It is true that this could have been expressed better in B6.1.16.
L135: calculated outputs - Is there a reference associated with this? Fine if not but useful if so.
Reference to Beven et al 1995 added

L149: both showed changes over time - I'm not clear, do you mean that the components have changed over time or that the represented time variable processes? Clarified

L151: later these stores were integrated – this would benefit from a reference. Added

L163: later versions of Topmodel – this would benefit from a reference. Done

L217: The BK79 paper … flashy catchment – this sentence doesn't make sense to me. Added wets and dries

L234: Whilst… - this sentence reads as a fragment. Corrected

L235: but were always clearly identified – I think this relates to the assumptions but it is not clear, perhaps there is a comma missing. Deleted

L256: including – should be included Corrected

L285: but t more – typo, remove t Corrected

L332-3: as the catchment wets and dries – An additional sentence explaining why would be helpful here Modified

L336-7: This should not be a surprise at Tarrawarra … impermeable subsoil – this would benefit from a reference Added

L346: connectivity of both surface and subsurface flows - It seems as though there are two issues being discussed here, disconnection and response timescale. 1) Even if the signal can propagate rapidly in the presence of a water table configured as assumed by Topmodel it may not propagate if that assumed configuration breaks down resulting partial or total disconnection of some upslope area. 2) Even if the water table is configured as assumed in topmodel the signal propagation may not be sufficient to generate steady state like response to rainfall. I thought Barling's modification was largely focussed on the second problem. That is correct - at that time the focus was on the response times – breakdown of subsurface connectivities came later. This has been made clearer

L348: This paragraph, and particularly this sentence contains two ideas that might be more easily understood if separated: subsurface and overland flow connectivity. This has been reordered and made clearer

L348: always connectivity – do you mean of subsurface flow? Added

L353: can be represented in… - It has also been represented in the network index, a modification to the topographic index to identify areas of disconnected saturation within Topmodel, the runoff from which is then assumed to entirely reinfiltrate as run-on (Lane et al., 2004, 2009, Lane and Milledge, 2013). References added

L355: wetter vegetation patterns – perhaps a reference here? Mostly personal observation but one reference added

L361: particularly – this was not clear to me. Do you mean that Kirkby shows it for the particular case of exponential transmissivity or that it is particularly true for the exponential transmissivity profile case? I think it is the latter. If so, what are the implications for other assumed transmissivity profiles? This paragraph has been deleted and incorporated in the more detailed discussion later

L365: parallel to the surface - How much empirical support is there for assumption A2? You provide a statement on this for A1 and a similar statement here would be useful. Wording changed. Also discussed more extensively later in the evaluation of predictions section

L366: important - It might be helpful to say important to what. This may be tricky given the wide range of applications of Topmodel and its derivatives and I don't think an exhaustive list of 'important to...' is necessary here. However, an indication that both the scale over which the assumption is not valid and the question being addressed contribute to whether or not violation of A2 will be important. For example, I could imagine bedrock exfiltration being very important to local patterns of saturation deficit on the scale of metres to tens of metres (and therefore to pore pressure at scales associated with landslide triggering) but much less important over scales of hundreds to thousands of metres and therefore for runoff generation. Text has been modified

L370: (it might… values) – Has anyone done this? If not it might be worth flagging that more strongly. One of the valuable things about this paper is the opinions of the original authors on what has been done and what is yet to be done from their perspectives. Yes, in the paper cited.

L374: values tend to be high - Is that a commensurability issue associated with scale dependence in the measurement? If so it might be useful to comment on that here. The following text already gives references to papers addressing this.

L388: A further criticism… unsaturated zone – a reference is needed here. Also, this doesn't seem to get the detailed response that other criticisms received. Is the following sentence a general conclusion to the section or specific to this criticism? Expanded

L425: small – adding the catchment area would be useful here. Added

L429-30: This was … number of storms – mixed tenses in this sentence   Corrected

L431: would predict – should this be: would predict (i.e. it is possible) or predicted (i.e. this was the prediction)?   Clarified

L463: predicted modelled – I think only one or the other of predicted or modelled is needed here. Corrected

L463: storage deficits to water tables - You don't give this much treatment. Can you point to a reference and give an indication of the magnitude of the issue, as written it appears a straightforward exercise.  This was already explained in L154ff but reference back added

L466: represent different quantities - An example here might be helpful. E.g. modelled water table depth at 2 m resolution compared with observations from a single well within that 2 m cell.  That is a bit difficult to illustrate without going into considerable depth.   We do refer to the Freer et al. which explains this in some detail.

L467: this is a particular problem - Is it commensurability between predictions and observations that is a particular problem in this case or commensurability between the input parameter (spatially uniform T0) and the local conditions (heterogeneous hydraulic conductivity in 3-D)?  Clarified

L470: would not expect the predictions to match… - Has anyone provided an indication of the likely magnitude of the commensurability errors? If so how do these compare to the misfit observed in these studies?  Several references already cited

L505: do not explicitly consider convergence… - I disagree: the indices of Barling, Woods, explicitly consider convergence etc through upslope area as the topographic index does, that of Berne uses a hillslope width function and that of Hjerdt was conceived as a replacement for local slope which has subsequently been used within a modified a/tanB topographic index.   Modified

L521: a rumour - this moves away from the authors' own experiences and is difficult to support. Consider rephrasing or removing the last two or three sentences since I think you can make your point without this statement.  Actually it was part of my experience at a Hydrocomp workshop with Crawford and Linsley in around 1974 and is part of the history.   Wording has been modified.

L528: that contributing – should be that the contributing   Modified

L601-5: One of the… A second result… - A reference that you associate with each of these results would be helpful at the end of each sentence.  References added

L605: longer time steps – could you quantify longer timesteps?  Changed to hourly

L626: risk - Could you expand this to replace 'risk'? You haven't previously defined risk and it may mean different things to different people. Changed to probability of exceedance

L646: A further step - Should this be 'an alternative step'? Dynamic Topmodel predates the distributed version and does not adopt its surface flow treatment. Now modified

L696-7: some interest - references to this interest would be useful here, unless the following sentence is an example. Modified

L729: topographic index - this actually uses the network index, a modified version of the topographic index that accounts for downslope connectivity of saturation excess overland flow (see: Lane et al., 2004; 2009). Corrected

L761: some of the other assumptions – it would be useful to be explicit about which assumptions here. Taken out since mostly routing and heterogeneity that are mentioned explicitly

L762: hydrological important – should be: hydrologically important. Corrected

L767: cease to allow – I think there is a comma missing here. Corrected

L780: This approach … much meaning – it is not clear to me what this comment means. Wording changed

At least, looking back on my own applications of Topmodel, I identify our inability to account properly for varying surface contributing area as a result of dynamic surface hydrological connectivity as a major bottleneck for our model performance (in addition to errors in the input

data). Even though this process could clearly be observed in the field, there was no way to incorporate this in the model (apart from some simple and unsatisfactory conceptual approaches) because of the lack of topographic information of sufficient quality. However, recent advances in drone-based remote sensing, includ- ing the generation of cm-resolution digital elevation models, may open interesting new opportunities in this regard and I would be very interested in hearing the authors' views on this.

This has been discussed (in the context of the past) in the section on connectivity in the paragraph L437.   High resolution topographic data is clearly more relevant to surface flow routing (see reference to sub-grid rills and small channels in that paragraph) than in determining flow pathways in the subsurface (also the discussion of Jim's work on bedrock topography L615.   Such issues will be important in any model and certainly reflected in effective values of parameters however calibrated.

(On the other end of topographic-data-availability-spectrum, I seem to recall some successful implementations of Topmodel that bypassed the topographic index derivation altogether in favour of the use of a gamma distribution with calibrated parameters.)   This results in more parameters to be estimated of course, but has now been incorporated into the text and into the timeline supplement

Specific comments
l55: there is an error in the formula (S on the right hand side should not be there) Corrected

l60: it may be useful to define $S_0$ here explicitly - also further in the document, not all symbols are always clearly defined   Done

l237: ofa -> of   Corrected

l249: attractive. -> attractive   Corrected

l285: "but t more"   Corrected

l452: "runoff." -> "runoff"   Corrected

l730: Lane et al. (2009) is referenced here in relation to SCIMAP although the publica- tion is not really about SCIMAP.   Also now properly refers to network index.

This reference has been added to the Suppplementary material.

[revised manuscript text omitted]